# Modulation of α-synuclein aggregation amid diverse environmental perturbation

Abdul Wasim, Sneha Menon, Jagannath Mondal*

Tata Institute of Fundamental Research, Hyderabad, India

**Abstract** Intrinsically disordered protein α-synuclein (αS) is implicated in Parkinson's disease due to its aberrant aggregation propensity. In a bid to identify the traits of its aggregation, here we computationally simulate the multi-chain association process of αS in aqueous as well as under diverse environmental perturbations. In particular, the aggregation of αS in aqueous and varied environmental condition led to marked concentration differences within protein aggregates, resembling liquid-liquid phase separation (LLPS). Both saline and crowded settings enhanced the LLPS propensity. However, the surface tension of αS droplet responds differently to crowders (entropy-driven) and salt (enthalpy-driven). Conformational analysis reveals that the IDP chains would adopt extended conformations within aggregates and would maintain mutually perpendicular orientations to minimize inter-chain electrostatic repulsions. The droplet stability is found to stem from a diminished intra-chain interactions in the C-terminal regions of αS, fostering inter-chain residue-residue interactions. Intriguingly, a graph theory analysis identifies *small-world-like networks* within droplets across environmental conditions, suggesting the prevalence of a consensus interaction patterns among the chains. Together these findings suggest a delicate balance between molecular grammar and environment-dependent nuanced aggregation behavior of αS.

*For correspondence:
jmondal@tifrh.res.in

Competing interest: The authors declare that no competing interests exist.

## eLife assessment

This study provides **important** biophysical insights into the molecular mechanism underlying the association of alpha-synuclein chains, which is essential for understanding the pathogenesis of Parkinson's disease. The data analysis is **solid**, and the methodology can help investigate other molecular processes involving intrinsically disordered proteins.

## Introduction

In the human body, a significant presence of intrinsically disordered proteins (IDPs) plays diverse and crucial roles (*Fuxreiter and Tompa, 2012*; *Forman-Kay and Mittag, 2013*; *Bah and Forman-Kay, 2016*). These proteins lack a well-defined 3D structure under native conditions, which imparts functional advantages, but also renders them susceptible to irreversible aggregation, especially when affected by mutations. Such aggregates can be pathogenic and are associated with various diseases, including neurodegenerative diseases, cancer, diabetes, and cardiovascular diseases (*Uversky et al., 2008*).

Notably, Alzheimer's disease is characterized by the aggregation of the amyloid-β peptide (Aβ), while Parkinson's disease (PD) is linked to α-synuclein (αS) aggregation. A growing body of evidence has established a connection between IDPs and the phenomenon known as *liquid-liquid phase separation* (LLPS). During LLPS, high and low concentrations of biomolecules coexist without the presence of membranes and exhibit properties similar to phase-separated liquid droplets of two immiscible liquids (*Figure 1*; *Xing et al., 2021*; *Ray et al., 2020*; *Shu et al., 2021*; *Rodríguez et al., 2023*; *Gui et al., 2023*). This intriguing phenomenon has garnered significant attention as it underlies the

**Figure 1.** A schematic showcasing the process of liquid-liquid phase separation of α-synuclein (αS).

formation of membrane-less subcellular compartments (*Hyman et al., 2014*; *Banani et al., 2017*; *Shin and Brangwynne, 2017*), which, when dysregulated, can lead to incurable pathogenic diseases.

Recent findings have highlighted the capability of αS to undergo LLPS under physiological conditions, specifically when the protein concentration surpasses a critical threshold (*Ray et al., 2020*). Moreover, it was observed that the aggregation propensity of αS is significantly influenced by various factors, including the presence of molecular crowders, the ionic strength of the protein environment, and pH (*Sawner et al., 2021*). Nonetheless, characterizing the interactions and dynamics of these small aggregates poses experimental challenges, leading to limited available reports on the subject (*Apetri et al., 2006*; *Hong et al., 2011*; *Chen et al., 2015*; *Cremades et al., 2017*).

This investigation aims to establish the molecular basis of self-aggregation of αS and underlying process of LLPS under diverse environmental perturbations. In particular, to understand the influence of environmental factors on the inter-protein interactions within a phase-separated droplet, we target to computationally simulate the aggregation process of αS under different conditions, emphasizing the roles of crowders and salt. While recent progress in computational force fields and hardware has enabled the simulation of individual IDPs especially αS, using all-atom molecular dynamics (AAMD) (*Ahmed et al., 2021*; *Bari and Prakashchand, 2021*; *Robustelli et al., 2018*; *Best et al., 2014*; *Huang et al., 2017*; *Menon and Mondal, 2023*; *Menon and Mondal, 2022*), these simulations can be extremely time-consuming and resource-intensive, making multi-chain AAMD simulations, even with cutting-edge software and hardware, impractical. Therefore, to simulate the the aggregation process, we resort to coarse-grained molecular dynamics (CGMD) simulations. Multiple CG force fields have been developed with the sole purpose of fast and accurate simulations of IDPs and LLPS (*Dignon et al., 2018*; *Regy et al., 2021*; *Joseph et al., 2021*; *Tesei and Lindorff-Larsen, 2022*). However these are implicit water, residue-level CG models. Therefore, here we leverage a tailored Martini 3 CG force field (CGFF) (*Souza et al., 2021*) for αS and use it to dissect the inter-protein interactions governing stable aggregate formation and LLPS. By leveraging the CGFF framework and building upon the groundwork laid by prior studies (*Benayad et al., 2021*; *Thomasen et al., 2022*; *Mukherjee et al., 2023*), we have optimized water-protein interactions for αS. Our multi-chain microsecond-long CGMD simulations have resulted in comprehensive ensembles of significant protein aggregates spanning various scenarios.

As one of the key observations, our simulation unequivocally reveals LLPS-like attributes in the aggregates and shows how these get modulated in the presence of crowders and salt. The investigation unearths the intricate interplay of mechanical and thermodynamic forces in αS aggregation, achieved through meticulous data analyses. We elucidate the pivotal intra- and inter-protein interactions governing LLPS-like protein droplet formation, unveiling the protein's primary sequence's role in aggregation. As would be shown in this article, a graph-based depiction of the droplet's architecture represents the proteins within droplets as constituting dense networks akin to *small-world networks*.

## Results

In this study, we utilized the recently developed Martini 3 (*Souza et al., 2021*) CG model to simulate collective interaction of a large number of αS chains in explicit presence of aqueous media at various

concentrations commensurate with in vitro conditions including the presence of crowders and salt. As Martini 3 was not originally developed for IDPs, we carefully optimized the protein-water interactions against atomistic simulation of monomer and dimer of αS, as detailed in the *Methods* section, to ensure compatibility with αS (see *Methods*).

Initially, we examined the impact of concentration on the protein's aggregation by simulating copies of chains, maintaining a polydispersity of protein conformations of αS. In particular, three different conformations of αS (referred to here as *ms*1, *ms*2, and *ms*3) with $R_g$s (radius of gyration) ranging between collapsed and extended states (1.84–5.72 nm) at different concentrations, with a composition, as estimated in a recent investigation (*Menon and Mondal, 2023*), were employed. First the chains were simulated for extensive period in a set of three protein concentrations, close to previous experiments.

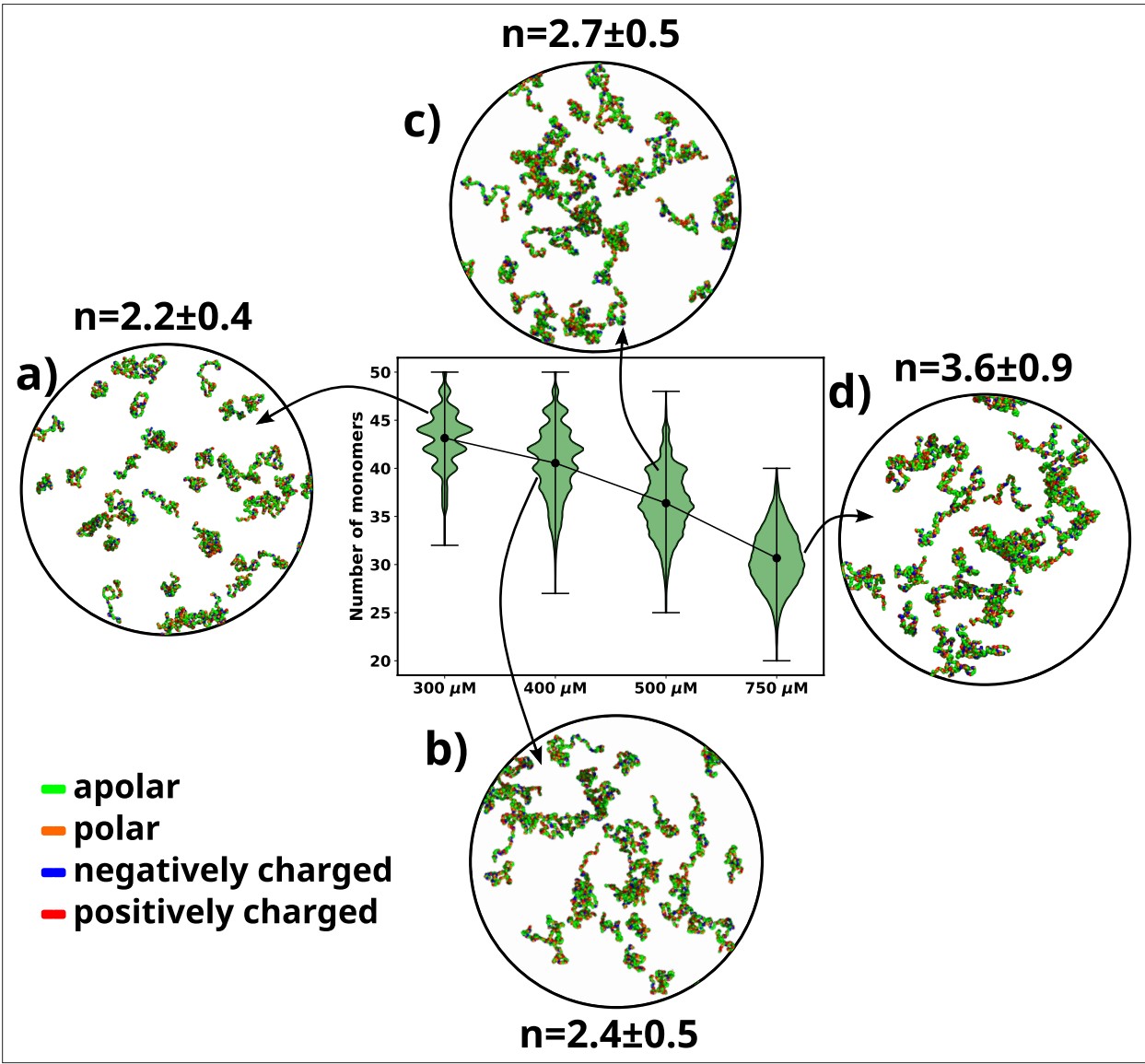

**Figure 2.** A violin plot showing the distribution of number of monomers present for different concentrations of α-syn. The blue dot at the middle of each distribution represents the mean number of monomers observed for each concentration. For each concentration we show representative snapshots of the system. For each concentration, we also report the statistics of the number of chains in the largest cluster (*n*). (**a**) A snapshot from the simulation at 300 μM α-syn. (**b**) A snapshot from the simulation at 400 μM α-syn. (**c**) A snapshot from the simulation for 500 μM α-syn. (**d**) A snapshot from the simulation at 750 μM α-syn.

## Simulations capture enhanced aggregation beyond a threshold concentrations of αS

We performed simulations of αS at various concentrations, namely 300 μM, 400 μM, 500 μM, and 750 μM. We begin by analyzing the aggregation behavior of αS. As shown in *Figure 2*, we observe that most chains do not aggregate at 300 and 400 μM as characterized by the prevalence of high number of free monomers. The respective snapshots of the simulation indicate the presence of greater extent of single chains. Also, the chains that are not free form very small oligomers of the order of dimer to tetramer (*Figure 2*).

However, upon increasing the concentration to 500 μM, which has also been the critical concentration reported for αS to undergo LLPS (*Ray et al., 2020*), we observe a sharp drop in the average number of free monomers in the system (*Figure 2*). The corresponding representative snapshot of the system also depicts a few higher-order aggregates, such as pentamers and hexamers, as well as most chains forming small oligomers. This can be understood from the value of the average number of chains present in the largest clusters, as reported in *Figure 2*.

The system, being at critical concentration, formation of large aggregates would require longer timescales than the simulation length. Therefore, in order to promote the formation of large aggregates (heptamers or more) for finer characterization, we performed a simulation at a higher concentration of 750 μM αS. As shown in *Figure 2d*, we observe further decrease in the total number of free monomeric chains in the solution. There is simultaneous appearance of a very few droplet-like aggregates (hexamer or more) as can be seen from *Figure 2* and the adjacent snapshot of the system (*Figure 2*). However, we note that ~60% of the protein chains are free and do not participate in aggregation and we think that as such in water, αS does not possess a strong and spontaneous self-aggregation tendency. In the following sections we characterize the aggregation tendency of αS in the presence of certain environmental modulator that can shed more light on this hypothesis.

## Molecular crowders and salt accelerate αS aggregation

The cellular environment, accommodating numerous biological macromolecules, poses a highly crowded space for proteins to fold and function (*Ellis and Minton, 2006*; *Deeds et al., 2007*; *Li et al., 2008*; *Zhou, 2013*). In in vitro studies, inert polymers such as Dextran, Ficoll, and polyethylene glycol (PEG) are commonly employed as macromolecular crowding agents. In the context of αS aggregation, previous experimental studies have revealed an increased rate of in vitro fibrillation in the presence of different crowding agents (*Uversky et al., 2002*; *Munishkina et al., 2008*; *Horvath et al., 2021*). Notably, a recent experimental study demonstrated the occurrence of phase separation (LLPS) of αS in the presence of PEG molecular crowder (*Ray et al., 2020*). Moreover, considering that in vivo environments also contain various moieties like salts and highly charged ions, a recent in vitro study has shown that the ionic strength of the solvent directly influences the aggregation rates of αS (*Sawner et al., 2021*), with higher ionic strength enhancing αS aggregation.

Given these observations, it becomes crucial to characterize the factors responsible for the enhanced aggregation of αS in the presence of crowders and salt. To address this, we perform two independent sets of simulations: one with αS present at 750 μM in the presence of 10% (vol/vol) fullerene-based crowders (see *SI Methods*) and the other with the same concentration of αS but in the presence of 50 mM of NaCl. In this section we characterize the effects of addition of crowders or salt on the aggregation of αS.

As expected, the addition of crowders leads to an enhancement of αS aggregation due to their excluded volume effects, as depicted in *Figure 3a*. Notably, the number of monomers drastically decreases upon the inclusion of crowders. This observation is further supported by the snapshots of the system, which also confirm the reduction in monomer count. Similarly, we observe that the presence of salt also promotes αS aggregation, as illustrated in *Figure 3a*, where the number of monomers is lower when compared to the case with no salt.

Following this, we conducted an analysis of the number of chains present in the largest clusters that formed. *Figure 3b* clearly illustrates that the addition of crowder or salt leads to a notable increase in the average number of proteins forming a cluster. This crucial observation points to the fact that the inclusion of accelerators, such as crowder or salt, not only promotes aggregation but also plays a role in stabilizing the formed oligomers. Importantly, we observed that the effect of crowder on aggregation is slightly more pronounced compared to that of the salt. In the subsequent section, we delve

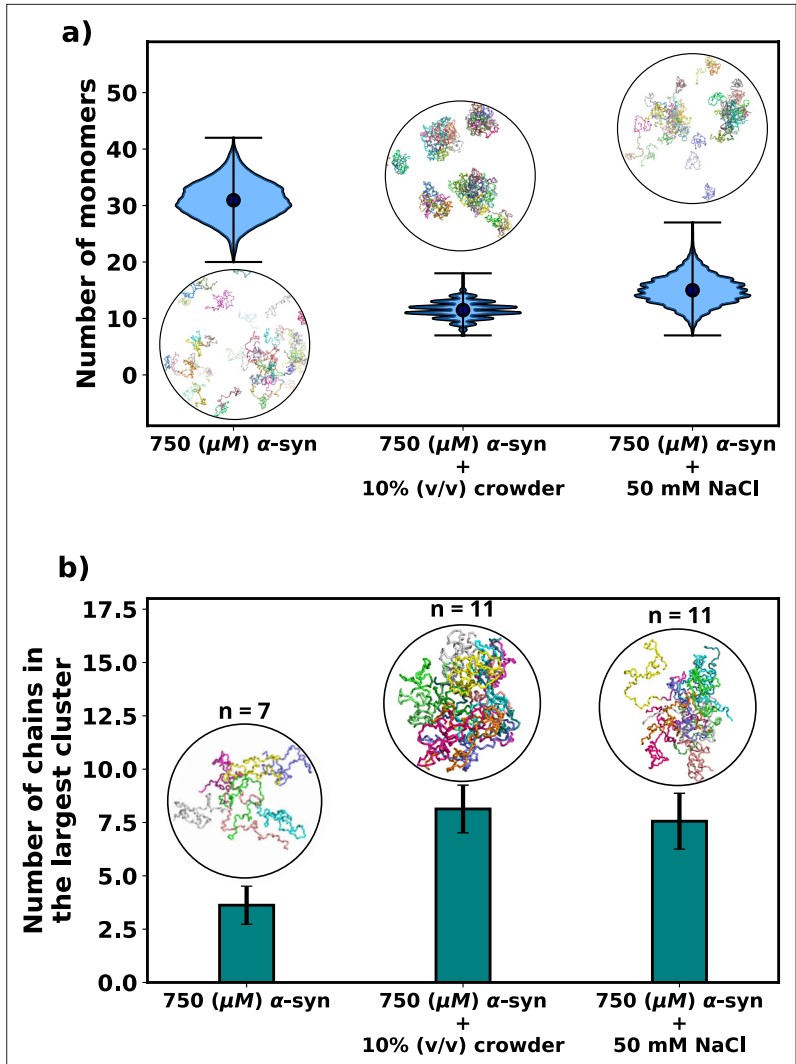

**Figure 3.** Effect of salt and crowder on αS aggregation. (**a**) A violin plot showing the distribution of the number of monomers for α-syn at 750 μM without and with crowder. The blue dots represent the means of each distribution. The snapshots represent the extent aggregation for a visual comparison. (**b**) A bar plot showing the number of chain in the largest cluster formed by α-syn at 750 μM without and with crowder. The snapshots show the largest cluster formed for each scenario.

into the reasons behind the enhanced aggregation induced by these accelerators, aiming to decipher the underlying mechanisms responsible for their influence on αS aggregation dynamics. As the aggregation is significant enough for performing quantitative analysis only when the concentration of αS is 750 μM, we perform all analysis on scenarios at 750 μM of αS.

## Crowders and salt differentially modulate surface tension for promoting LLPS-like αS droplets

The preceding sections underscore our simulation-based observation that, influenced by crowders and salt, αS aggregates into higher-order oligomers (hexamers and beyond) at a significantly accelerated propensity compared to the scenario without these influences. Here, we delve into the investigation of the energetic aspects underlying this aggregation phenomenon. An important contributor to the energetics is the surface tension, arising from the creation of interfaces between the dense and dilute phases of the protein upon droplet formation. This presence of interfaces is accompanied by surface tension and surface energy. The surface energy of a system is directly proportional to its surface area; systems with higher surface energy tend to minimize their surface area. Consequently,

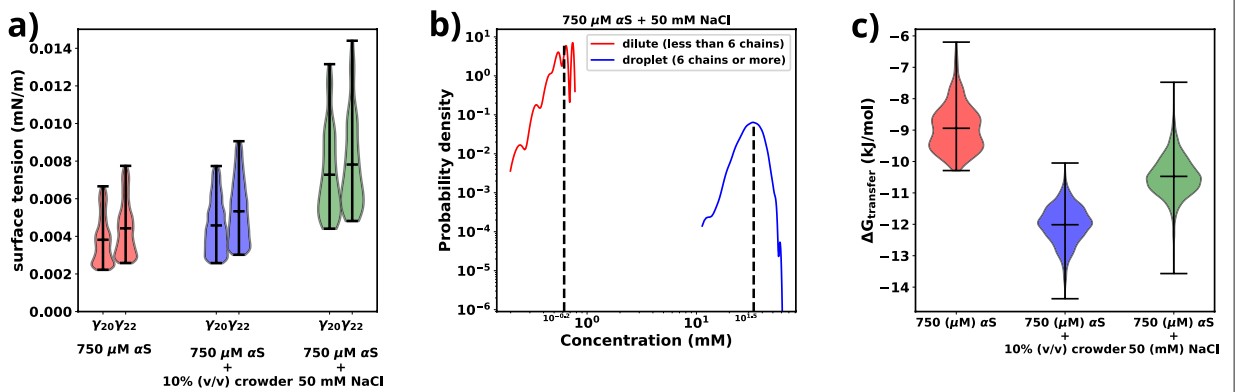

**Figure 4.** Exploring energetics of αS aggregation. (**a**) Surface tensions of droplets, estimated from $\gamma_{20}$ and $\gamma_{22}$, for three cases have been shown. Both $\gamma_{20}$ and $\gamma_{22}$ provide almost similar estimates of the value of surface. (**b**) Comparison of protein concentrations for the dilute (red) and the droplet (blue) phases for 750 µM αS+50 mM NaCl. (**c**) Excess free energy of transfer comparison for three cases.

The online version of this article includes the following figure supplement(s) for figure 4:

**Figure supplement 1.** Comparison of protein concentration at two different phase.

systems comprising multiple smaller droplets exhibit a larger surface area, and hence a higher surface energy. Conversely, systems characterized by fewer, larger droplets possess a comparatively reduced surface area and correspondingly lower surface energy. This insight leads us to conjecture that surface tension could play a pivotal role in driving LLPS and the formation of larger αS droplets. To explore this hypothesis, we calculate the surface tension of the resultant droplets, as per **Equations 1 and 2** and as described in *SI Methods* and **Benayad et al., 2021**.

$$\gamma_{20} = \frac{5k_BT}{16\pi\langle(\delta a + \delta b)^2\rangle} \tag{1}$$

$$\gamma_{22} = \frac{15k_BT}{16\pi\langle(\delta a - \delta b)^2\rangle} \tag{2}$$

where $\delta a = a - R$ and $\delta b = b - R$ is the perturbation of the droplet shape from a perfect sphere with a radius $R$ along any two pairs of principle axes of general ellipsoid estimating the shape of the droplet. The surface tension ($\gamma$) is thus estimated using $\gamma \approx \gamma_{20} \approx \gamma_{22}$. Please see *SI Methods* and **Benayad et al., 2021**, for more details.

**Figure 4a** provides a comparison of the surface tension ($\gamma$), for three different scenarios involving αS: (i) αS in solution, (ii) αS in the presence of crowders, and (iii) αS in the presence of salt. Notably, in each case, the surface tension is considerably lower (0.0035–0.0075 mN/m) than the surface tension for FUS droplets in water (~0.05 mN/m) (**Benayad et al., 2021**). As stated earlier, the magnitude of surface tension is an estimate of the aggregation tendency of any liquid-liquid mixture. Since we find that $\gamma_{\alpha S}$ is much lower than $\gamma_{FUS}$, we assert that the propensity with which αS aggregates should be much lower than that of FUS.

Next, we conduct a comparison of the three different scenarios to understand the effects of crowders and salt on the aggregation of αS. From **Figure 4a**, it is evident that the surface tensions are very similar for cases (i) and (ii), while it has increased for case (iii). This implies that the addition of crowders does not significantly impact the surface tension of the aggregates, although it renders the protein more prone to aggregation. On the other hand, the addition of salt causes an increase in surface tension. Given the relationship between surface area and volume, where a higher surface-to-volume ratio signifies numerous smaller droplets, the surface energy is concurrently elevated. In the presence of salt, a tendency is observed for these smaller aggregates to coalesce, giving rise to larger aggregates, albeit in reduced numbers. This behavior is an endeavor to curtail the surface-to-volume ratio and thus mitigate the associated surface energy. Therefore, the larger the surface tension, the higher is tendency of the protein to form aggregates, as seen from the surface tension values of αS and FUS, as mentioned earlier.

To minimize the surface energy, fusion of aggregates, either via merging of two or more droplets into one is seen for liquid-like phase-separated droplets in experiments (*Ray et al., 2020*). Although droplet fusion was not observed in our simulations due to the limited system size, it was shown that if a protein undergoes LLPS, a significant difference in protein concentration occurs between the droplet and the dilute phase (*Nguyen et al., 2022*). To verify whether the aggregates observed in our simulations exhibit characteristics of LLPS, we calculated the protein concentrations in the dilute and concentrated phases. For the droplet phase, the concentration of the protein was calculated using *Equation 3*.

$$c_{phase} = \frac{N_{phase}}{N_A \times V_{phase}}$$ (3)

where $N_{phase}$ is the number of protein chains in the phase (here dilute or concentrated), $N_A$ is Avogadro's number, and $V_{phase}$ is the volume occupied by the phase. For the dilute phase, we estimated the volume of the concentrated/dense phase ($V_{dense}$) using *Equation 4* (*Nguyen et al., 2022*).

$$V^i_{dense} = 4\pi\sqrt{3}\lambda_1\lambda_2\lambda_3$$ (4)

where $V^i_{dense}$ is the volume of the $i$th droplet, $\lambda_1$, $\lambda_2$, and $\lambda_3$ are the eigenvalues of the gyration tensor for the aggregate. The volume of the dilute phase is the remainder volume of the system given by *Equation 5*.

$$V_{dilute} = V - \sum_i V^i_{dense}$$ (5)

where $V$ is the total volume of the system.

As shown in *Figure 4b* and *Figure 4—figure supplement 1*, there is an almost two orders of magnitude difference between the concentration of αS in the dilute and droplet phases for all scenarios. Such a pronounced difference is a hallmark of LLPS, leading us to assert that the aggregates formed in our simulations possess LLPS-like properties. Consequently, we use the term 'droplet' interchangeably with 'aggregates' for the remainder of our investigation.

$$\Delta G_{transfer} = RT \ln\left(\frac{c_{dilute}}{c_{dense}}\right)$$ (6)

Finally, utilizing the calculated concentrations, we proceed to estimate the excess free energy of monomer transfer ($\Delta G_{transfer}$), from *Equation 6*, between the dilute and droplet phases, where $c_{dilute}$ is the concentration of αS in the dilute phase, $c_{dense}$ is the concentration of αS in the dense/droplet phase, $R$ is the universal gas constant, and $T$ is the temperature of the system (=310.15 K). As illustrated in *Figure 4c*, both crowder and salt scenarios demonstrate lower $\Delta G_{transfer}$ values compared to the case without their presence. However, the thermodynamic origins behind this pronounced aggregation differ for crowders and salt. Crowders enhance aggregation primarily through excluded volume interactions, which are of an entropic nature. On the other hand, salt enhances aggregation by increasing the droplet's surface tension, thus contributing to the enthalpy of the system. As a result, apart from the already known fact that macromolecular crowding decreases $\Delta G_{transfer}$ via entropic means, we also infer that salt decreases $\Delta G_{transfer}$ via enthalpic means by increasing the surface tension of the formed droplets.

## Aggregation results in chain expansion and chain reorientation in αS

An indicative trait of molecules undergoing LLPS is the adoption of extended conformations upon integration into a droplet structure. Given that the aggregates observed in our simulations exhibit a concentration disparity reminiscent of LLPS between the dilute and dense phases, we endeavored to validate the presence of a comparable chain extension phenomenon within our simulations (*Nguyen et al., 2022*). To address this, we quantified the radius of gyration ($R_g$) for individual chains and classified them based on whether they were situated in the dilute or dense phase. The distribution of $R_g$ values for each category is illustrated in *Figure 5a* and *Figure 5—figure supplement 1*. Remarkably,

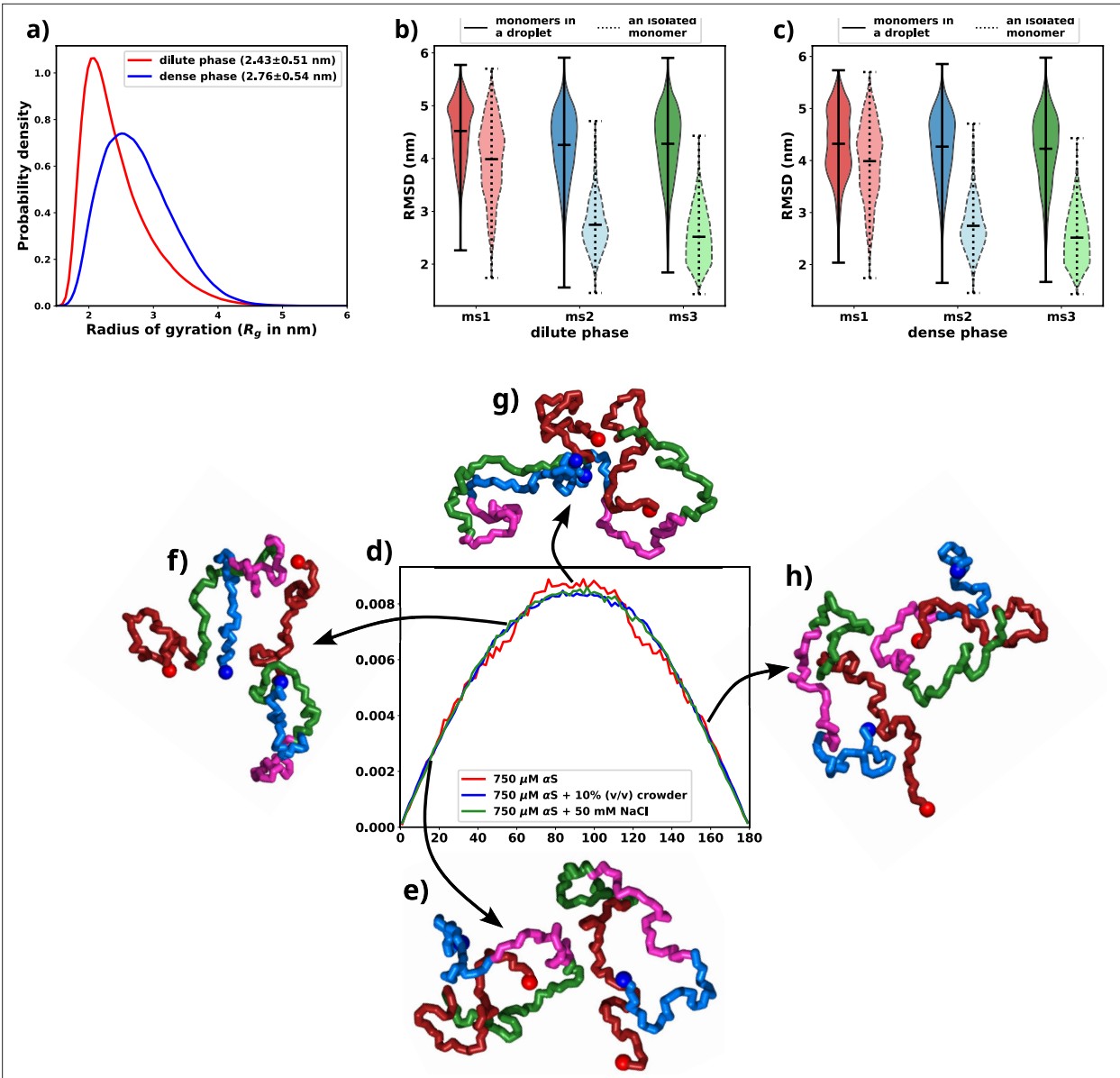

**Figure 5.** Exploring conformational change in αS monomers upon LLPS. All the figures are for 750 µM αS+50 mM NaCl. (**a**) Distribution of $R_g$ for proteins present in the dense or the dilute phases. (**b**) Comparison of root mean square deviation (RMSD) for protein chains present in the dilute phase, with single-chain RMSDs as the reference (dotted edges). (**c**) Comparison of RMSD for protein chains present in the dense phase, with single-chain RMSDs as the reference (dotted edges). (**d**) Distribution of the angle of orientation of two chains inside the droplet for the three different scenarios. (**e**) Representative snapshot for angle between 0 and 20 degree. (**f**) Representative snapshot for angle between 50 and 70 degree. (**g**) Representative snapshot for angle between 80 and 120 degree. (**h**) Representative snapshot for angle between 150 and 180 degree.

The online version of this article includes the following figure supplement(s) for figure 5:

**Figure supplement 1.** Comparison of monomer size in in liquid and dense phase.

the distribution associated with the dense phase distinctly indicates that the protein assumes an extended conformation within this context. As elucidated earlier, this marked propensity for extended conformations aligns with a characteristic hallmark of LLPS as previously seen in experiments (***Ubbiali et al., 2022***).

Having observed the conformational alterations of αS during LLPS, our subsequent aim was to quantify the extent of these conformational changes in relation to their initial states (referred to as 'ms1', 'ms2', or 'ms3' in decreasing order of $R_g$; ***Menon and Mondal, 2023***). To achieve this, we computed the root mean square deviation (RMSD) of each protein relative to its starting conformation.

The resulting distributions were visually depicted using violin plots, featuring bold edges in **Figure 5b and c**. The protein ensemble was segregated into two categories: (i) those from the dilute phase (**Figure 5b**) and (ii) those from the dense phase (**Figure 5c**).

Surprisingly, regardless of their initial configurations, the observed RMSD values were notably high. To facilitate a comparative analysis, we also included distributions of RMSDs for single chains simulated in the presence of 50 mM of salt, depicted using violin plots with broken edges. Intriguingly, the conformational state labeled as ms1, exhibited the least RMSD, a characteristic attributed to its notably extended conformation. This phenomenon aligns with the preference of droplets for extended conformations, implying that ms1 required the least conformational perturbation and thus exhibited a lower RMSD.

For both ms2 and ms3, a conspicuous increase in RMSD values was observed across all proteins monomers, irrespective of their respective phases. This phenomenon can potentially be attributed to the pronounced conformational shift experienced by the protein during aggregation. Building on these observations, we put forward a hypothesis: LLPS engenders significant modifications in the native protein conformations, ultimately favoring the adoption of extended states.

As discussed in the previous paragraph that the αS monomers inside the droplets must undergo conformational expansion and we hypothesized that they adopt orientation so as to minimize the inter-chain electrostatic repulsions. To this end, we try to decipher the orientations of the chains via defining their axes of orientations and subsequently calculating the angles between the major axis of two monomers. We calculate the major axis of gyration, given by the eigenvector corresponding to the largest eigenvalue of the gyration tensor, for each monomer inside a droplet. We next find the nearest neighbor (minimum distance of approach <8 Å) for each monomer, carefully taking care of over-counting.

The angle between two monomers is defined as the angle between the major axes of gyration between chain $i$ and its nearest neighbor $j$. We plot the distributions of the angles for all scenarios and all droplets in **Figure 5d**. We observe that irrespective of the conditions, the distribution peaks at right angles. The representative snapshots (**Figure 5e–h**) showcase their mode of orientation. Interestingly the distribution is the same for all the three scenarios, again stressing upon the fact the αS droplets share similar features in terms of interactions and orientations irrespective of their environments.

## Characterization of molecular interactions in aggregation-prone conditions

As established in preceding sections, both crowders and salt have been observed to augment the aggregation of αS while concurrently stabilizing the resultant aggregates. This phenomenon leads to

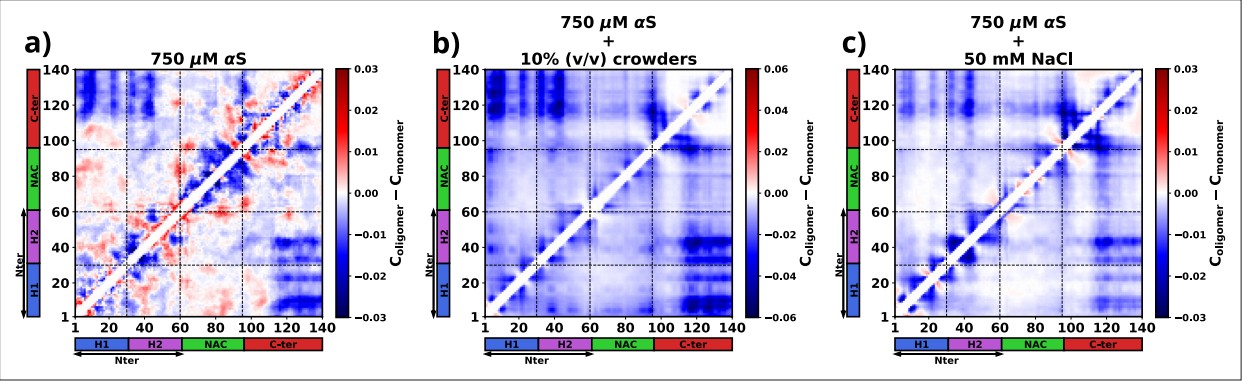

**Figure 6.** The figure presents the residue-wise, intra-protein difference contact maps where the average contact probability of monomers in the dilute phase was subtracted from the average contact probability of monomers in the dense/droplet phase for three cases. (**a**) 750 μM α-synuclein (αS) in water. (**b**) 750 μM αS in the presence of 10% (vol/vol) crowders. (**c**) 750 μM αS in the presence of 50 mM NaCl.

The online version of this article includes the following figure supplement(s) for figure 6:

**Figure supplement 1.** The intra-protein contact probability heatmap for proteins in the dilute phase for three scenarios.

**Figure supplement 2.** The inter-protein contact probability heatmap for proteins in the dense phase for three scenarios.

**Figure supplement 3.** The difference in inter-protein contact probabilities heatmap for proteins in the dense phase.

the protein adopting extended conformations within a notably heterogeneous ensemble. Shifting our attention, we now delve into a residue-level investigation to unravel the specific interactions responsible for stabilizing these aggregates and, consequently, facilitating the aggregation process.

To compute the differential contact maps, our approach involved initial calculations of average intra-protein residue-wise contact maps, termed as intra-protein contact probability maps, for monomers present in both the dilute and dense phases (refer to *Figure 6—figure supplement 1*). Subsequently, we derived the difference by subtracting the contact probabilities of monomers within the dilute phase from those within the dense phase. As evident from *Figure 6a*, a discernible reduction in intra-chain Nter-Cter interactions is observed for monomers within the droplet phase, depicted by the presence of blue regions along the off-diagonals. Such a reduction in such interactions has also been observed via experiments (*Ubbiali et al., 2022*) and it is similarly noticeable in the two other cases, as evident in *Figure 6b and c*.

Furthermore, a significant decline in intra-protein interactions, especially the NAC-NAC interactions, is predominantly observed at shorter ranges, indicated by deep-blue regions concentrated near the diagonals. Notably, these diminished intra-chain interactions (*Figure 6* and *Figure 6—figure supplement 1*) potentially facilitate the formation of inter-chain interactions (*Figure 6—figure supplement 2*). Thus, we observed that increased inter-chain NAC-NAC regions (*Figure 6—figure supplement 3*) facilitate the formation of αS droplets which also have been previously seen from FRET experiments on αS LLPS droplets (*Ray et al., 2020*). Building on these observations, we posit that these interactions play a pivotal role in stabilizing the aggregates that have formed.

Moreover, from the difference heatmaps in Appendix 1—figure 5, it can be observed that the residues 95–110 (VKKDQLGKNEEGAPQE) have reduced contact probabilities upon introduction of crowders/salt, whereas the rest of the contacts have slightly increased. These residues are highly charged and we think that upon introduction of crowders/salt, the proteins inside the droplet needed to be spatially oriented to facilitate the formation of largest aggregates. This re-orientation occurs to minimize the electrostatic repulsions among these residues belonging to different chains. These analyses provide hints that these residues are present in the protein so as to avoid the formation of aggregation-prone conformations, which is why their interactions had to be minimized to form more stable and larger aggregates.

## Phase-separated αS monomer forms *small-world* networks

The investigations so far suggest that irrespective of the factors that cause the aggregation of αS, the interactions that drive the formation of droplet remain essentially the same. However, the conformations of the monomers vary depending on their environment. In the presence of crowders they adapt to form much more compact aggregates. Therefore, here we characterize whether the environment influences the connectivity among different chains of the protein inside a droplet.

*Figure 7a, b, and c* shows molecular representations of the largest cluster formed by αS at 750 μM in water, αS at 750 μM in the presence of 10% (vol/vol) crowders, and αS at 750 μM in the presence of 50 mM NaCl, respectively. From the molecular representations for aggregates, it can be seen that irrespective of the system, they form a dense network whose characterization is not possible directly. Therefore, we represent each aggregate as a graph with multiple nodes (vertices) and connections (edges), as can be seen from *Figure 7d, e, and f*. Each node (in blue) represents a monomer in the droplet. Two nodes have an edge (line connecting two nodes) if the minimum distance of approach of the monomers corresponding to the pair of nodes is at least 8. We can see from the graph that not all chains are in contact with each other. They rather form a relay where a few monomers connect (interact) with most of the other protein chains. The rest of the chains have indirect connections via those. Since inter-chain connections/interactions have been denoted by edges and the chains themselves as nodes, such form of inter-chain interactions inside a droplet lead to only a few nodes having a lot of edges, e.g., node 1 in *Figure 7f*. The rest of them have only a few (3–5) edges. This is a signature of *small-world networks* (*Watts and Strogatz, 1998*; *Barrat and Weigt, 2000*; *Humphries and Gurney, 2008*; *Farag et al., 2022*) and we assert that αS inside the droplet(s) form small-world-like networks.

A network can be classified as a small-world network by calculating the *clustering coefficient* and the average shortest path length for the network and comparing those to an equivalent *Erdos-Renyi* network (*Rényi, 1959*). The *clustering coefficient* (*C*) is a measure of the 'connectedness' of a graph,

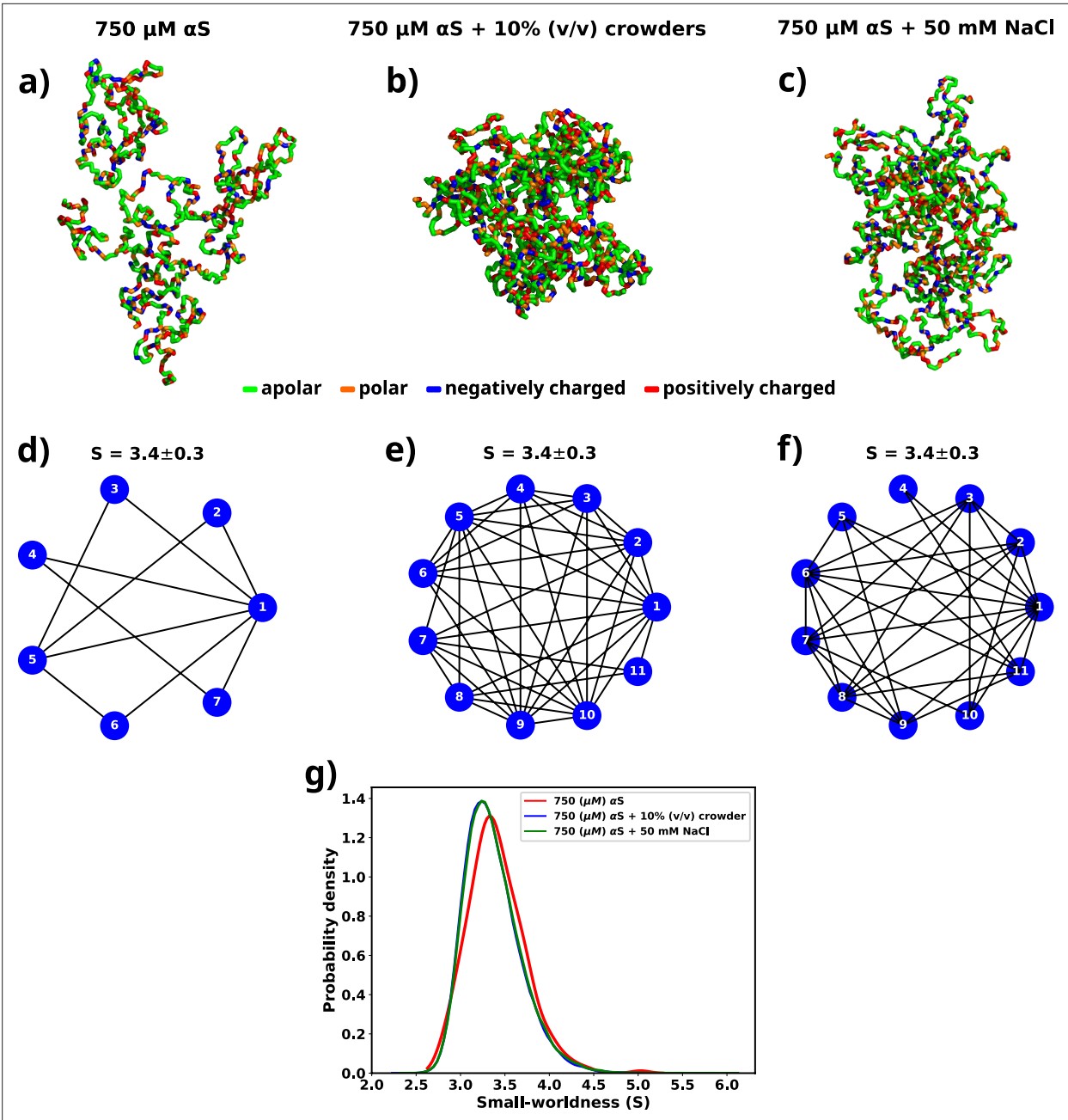

**Figure 7.** A graph theoretic analysis to characterize the "connectedness" of aS chains inside a droplet. (**a**) The largest cluster formed by αS at 750 μM. (**b**) The largest cluster formed by αS at 750 μM in the presence of 10% (vol/vol) crowder. (**c**) The largest cluster formed by αS at 750 μM in the presence of 50 mM salt. Different residues have been color coded as per the figure legend. (**d**) A graph showing the contacts among different chains constituting the largest cluster formed by αS at 750 μM. (**e**) A graph showing the contacts among different chains constituting the largest cluster formed by αS at 750 μM in the presence of 10% (vol/vol) crowder. (**f**) A graph showing the contacts among different chains constituting the largest cluster formed by αS at 750 μM in the presence of 50 mM NaCl. The mean small-worldness (*S*) of all droplet has been reported above the graph. (**g**) Distribution of small-worldness (*S*) for all scenarios.

indicating the extent to which nodes tend to cluster together. It quantifies the likelihood that two nodes with a common neighbor are also connected. On the other hand, the *average shortest path length* (*L*) is a metric that calculates the average number of steps required to traverse from one node to another within a network. It provides a measure of the efficiency of information or influence propagation across the graph. To estimate the *small-worldness* of a graph, we calculate a parameter (*S*) defined by *Equation 7*.

$$S = \frac{\frac{C}{C_r}}{\frac{L}{L_r}} \tag{7}$$

where $C$ and $L$ are the clustering coefficient and average shortest path length for the graph generated for a droplet respectively, while $C_r$ and $L_r$ clustering coefficient and average shortest path length for an equivalent Erdos-Renyi network, respectively. Small-world networks exhibit the characteristic property of having $C >> C_r$, while $L \approx L_r$. In light of this, for every scenario (solely αS, αS in the presence of crowder, and αS in the presence of salt), we generate an ensemble of graphs that correspond to the droplets formed during the simulation.

For each graph, we calculate the small-worldness coefficient ($S$) (*Humphries and Gurney, 2008*) and illustrate the distribution in *Figure 7g*. We observe a narrow distribution of $S$ with a mean of 3.4 for all cases. In a previous report of RNA-LLPS, a value of $S \approx 4$ was used to classify the droplets small-world networks (*Nguyen et al., 2022*). Therefore, $S = 3.4$ would suggest that the droplets formed during the simulations are small-world like. Moreover, we observe that the distribution of $S$ is invariant with respect to the environment of the droplet.

Therefore, we establish that the modes of interactions, orientations, and even connectivities among αS monomers inside a droplet remain same even when their environments are extremely different. We think that this occurs since the residue-level interactions among different monomers inside the droplet are similar irrespective of the environment, as shown in a previous section. This puts forth a very interesting way of viewing αS LLPS. We think that if these residue-level interactions can be disturbed then the stability of the formed droplets might be affected in such a way that they might dissolve spontaneously.

## Discussion

We used simulations to investigate the molecular basis of αS monomeric aggregation into soluble oligomers resembling micro-LLPS. The WT protein demonstrated limited aggregation, suggesting a low inherent propensity for LLPS dictated by its primary sequence. IDPs, like αS, often share primary sequence characteristics associated with phase separation. Charged residues distributed with uncharged amino acids, resembling the 'sticker and spacer' model, contribute to this molecular grammar. This observation aligns with a general trend in IDPs (*Choi et al., 2019*; *Martin et al., 2020*; *Choi et al., 2020*). To assess αS LLPS propensity from its primary sequence, we calculated Shannon entropy ($S$) (*Shannon, 1948*, *Equation 8* and *Figure 8—source data 1*), Kyte-Doolittle hydrophobicity (*Kyte and Doolittle, 1982*; *Figure 8—source data 2*), normalized, maximum of the sum of PLAAC log-likelihood ratios (NLLR) (*Lancaster et al., 2014*; *Figure 8—source data 3*), and LLPS propensity scores obtained from catGranules webserver (*Bolognesi et al., 2016*; *Figure 8—source data 4*).

$$S = \sum_i p_i \log p_i \tag{8}$$

where $p_i$ is the probability of occurrence of a residue in a given sequence.

Comparative analysis with three datasets (*Saar et al., 2021*), namely LLPS+: a dataset of high propensity IDPs whose critical concentrations are 100 µM or below, LLPS-: a dataset of low propensity IDPs whose critical concentrations are greater than 100 µM, and PDB*: a dataset of folded proteins that do not undergo LLPS under normal conditions, revealed αS's distinctive features (*Figure 8—source data 5*).

We note a significant difference in the Shannon entropy value of αS compared to proteins that do not undergo phase separation, as illustrated in *Figure 8a*. This deviation suggests a notable inclination of αS to undergo phase separation (*Saar et al., 2021*). Additionally, the hydrophobicity of αS (*Figure 8b*) is lower than that of the PDB* dataset, aligning more closely with the upper extremes of the LLPS- dataset. This indicates that while αS exhibits a tendency to undergo phase separation, the propensity should be low. Consistent with this, NLLR scores obtained from PLAAC and LLPS propensity scores (*Figure 8c and d*) reinforce this observation. These collective comparisons, coupled with simulations and experimental data on its critical concentration (*Ray et al., 2020*), conclusively establish that αS does not possess a high LLPS-forming propensity. Instead, this behavior is inherent to its

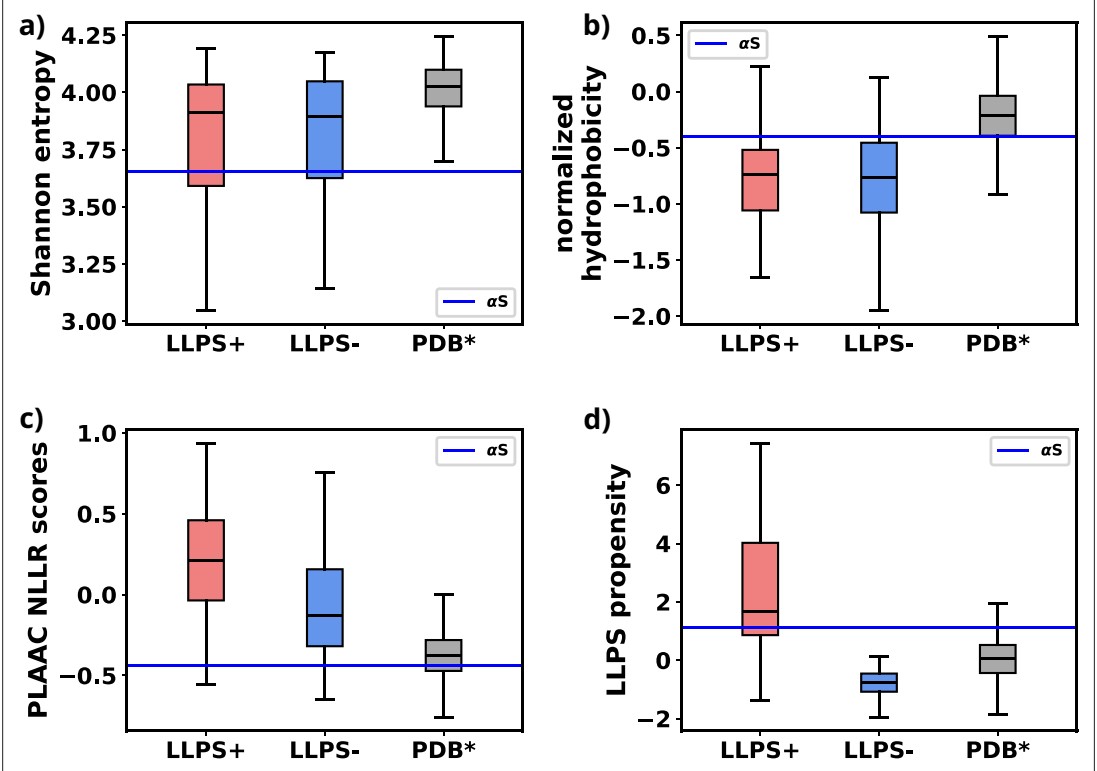

**Figure 8.** Comparison of primary sequence derived features for various datasets and aS. (**a**) Comparison of Shannon entropy of different datasets with αS. (**b**) Comparison of Kyte-Doolittle hydrophobicity of different datasets with αS. (**c**) Comparison of LLR scores, obtained from PLAAC, of different datasets with αS. (**d**) Comparison of liquid-liquid phase separation (LLPS) propensity scores, obtained from catGRANULE webserver, of different datasets with αS. The values have been summarized in *Figure 8—source data 5*.

The online version of this article includes the following source data for figure 8:

**Source data 1.** Shannon entropy (*Shannon, 1948*) for various datasets and α-synuclein (αS).

**Source data 2.** Normalized Kyte-Doolittle hydrophobicity (*Kyte and Doolittle, 1982*) scores for various datasets and α-synuclein (αS).

**Source data 3.** PLAAC normalized, maximum of the sum of PLAAC log-likelihood ratios (NLLR) (*Lancaster et al., 2014*) scores for various datasets and α-synuclein (αS).

**Source data 4.** catGRANULE (*Bolognesi et al., 2016*) scores for various datasets and α-synuclein (αS).

**Source data 5.** Comparison of primary sequence derived features for various datasets and α-synuclein (αS).

primary structure. In hindsights, this analysis also justifies the requirements of environmental factors for enhancing the proclivity of αS for LLPS, as demonstrated in both our simulations and experimental findings (*Ray et al., 2020*; *Sawner et al., 2021*).

For characterizing αS's aggregation phenomena, we calculated droplet surface tension under varied conditions. We observed that crowders minimally impacted surface tension, while salt increased it; however, both scenarios decreased the relative free energy of the system. Crowders achieved this via entropic means, whereas salt employed enthalpic means. Residue-residue interactions during droplet formation were consistent across environments, with crowder or salt enhancing these interactions. The aggregation pathway involved overall inter-chain interaction enhancement, specifically reducing intra-chain Nter-Cter and Nter-NAC interactions, leading to more extended protein conformations in droplets. The comparison with reported FRET observations (*Ray et al., 2020*) aligns well with the findings from our simulations, indicating that within the droplets, intra-chain NAC-NAC interactions have been supplanted by inter-chain NAC-NAC interactions. Droplet proteins displayed consistent orientation and 'small-worldness', a measure of inter-chain connectivity, remained consistent across diverse conditions. Thus, αS aggregates appeared invariant regarding their initial environment in terms of interactions and contacts.

Our study's precision was notably influenced by the careful selection of a simulation force field. Despite the availability of modern force fields optimized for multi-chain simulations of IDPs (*Dignon et al., 2018*; *Latham and Zhang, 2020*; *Regy et al., 2021*; *Zhang et al., 2022*), we opted for Martini 3, an explicit water model, due to its emphasis on water's role in aggregation and LLPS, as recently demonstrated in FUS LLPS (*Mukherjee and Schäfer, 2023*). Although newer models operate at a faster pace, Martini 3's inclusion of explicit water enhances result accuracy. Additionally, Martini 3 provides a detailed amino acid description and allowing for encoding of protein secondary structures, unlike some newer models that represent amino acids as single beads. Our meticulous choice of the simulation model, combined with a comprehensive analysis, contributes to the accuracy and novelty of this study.

Recent studies have explored the aggregation and LLPS of biopolymers and polyelectrolytes in the presence of membranes, opening a promising avenue for αS research (*Mondal and Cui, 2022*; *Liu et al., 2023a*; *Liu et al., 2023b*). Given that under physiological conditions, αS assumes an oligomeric, membrane-bound form, investigating its interactions with membranes could hold therapeutic potential (*Pineda and Burré, 2017*).

Under physiological conditions, crowding effects emerge prominently. While crowders are commonly perceived to be inert, as has been considered in this investigation, the morphology, dimensions, and chemical interactions of crowding agents with αS in both dilute and dense phases may potentially exert considerable influence on its LLPS. Hence, a comprehensive understanding through systematic exploration is another avenue that warrants extensive investigation.

Although we exclusively focused on wild-type αS, familial mutations have been reported to exhibit a significantly higher propensity for aggregation (*Ray et al., 2020*). These mutations, involving minor alterations in the primary sequence, highlight the importance of understanding the molecular basis of this distinctive phenotype. Additionally, the observed stability of pre-formed αS droplets (*Uversky et al., 2001*) poses a challenge in treating PD. Reversing aggregation/LLPS and understanding associated pathways and mechanisms are crucial. Our study identifies key residues crucial for stable droplet formation, consistent across various environmental conditions.

The significance of the solvent in αS aggregation remains underexplored. Recent studies (*Benayad et al., 2021*; *Mukherjee and Schäfer, 2023*) underscore the pivotal role of water as a solvent in LLPS. It suggests that comprehending the solvent's role, particularly water, is essential for attaining a deeper grasp of the thermodynamic and physical aspects of αS LLPS and aggregation. By delving into the solvent's contribution, researchers can uncover additional factors influencing αS aggregation. Such insights hold the potential to advance our comprehension of protein aggregation phenomena, crucial for devising strategies to address diseases linked to protein misfolding and aggregation, notably PD. Future investigations focusing on elucidating the interplay between αS, solvent (especially water), and other environmental elements could yield valuable insights into the mechanisms underlying LLPS and aggregation. Ultimately, this could aid in the development of therapeutic interventions or preventive measures for Parkinson's and related diseases.

## Methods
### Selection of the metric for optimizing water-protein interactions

We have opted to utilize the radius of gyration ($R_g$) of αS as the primary metric for optimizing water-protein interactions in Martini 3 for αS. To calibrate the Martini 3 force field, we employed 73 μs of all-atom data obtained from DE Shaw Research. From a polymer physics perspective, modifying water-protein interactions entails altering the solvent characteristics surrounding the biopolymer. We believe that $R_g$ serves as an effective metric in this context. Additionally, we focus on matching the distribution of $R_g$ values rather than solely targeting the mean value. This approach implies that, at a molecular level, the CGMD simulations conducted with optimized water-protein interactions enable the protein to explore conformations present in the all-atom ensemble.

Furthermore, we conducted cross-validation by comparing the fraction of bound states in all-atom and CGMD dimer simulations. This we claim that $R_g$ is good metric to be used for tuning of water-protein interactions in Martini 3.

### Optimizing Martini 3 parameters for αS

Martini 3 (*Souza et al., 2021*) was trained using DES-Amber (*Piana et al., 2020*) that is an atomistic force field tuned for single-domain and multi-domain proteins. Therefore, the default parameters of

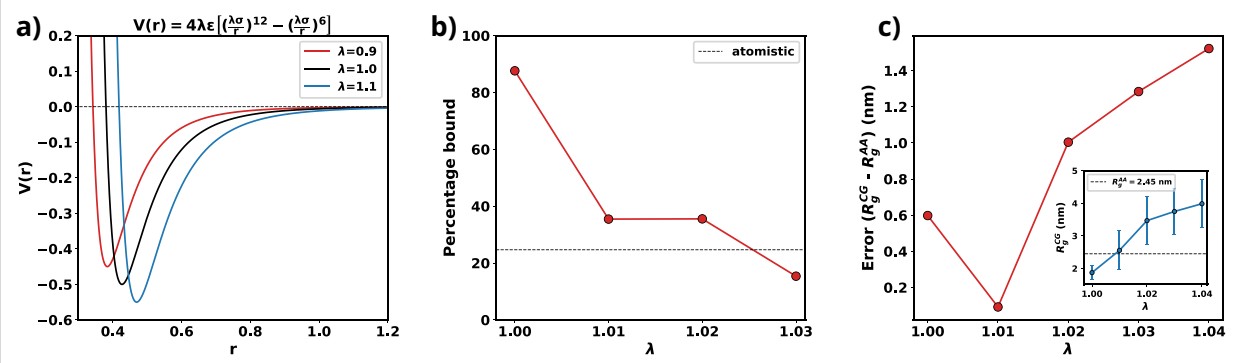

**Figure 9.** Optimization of Martini 3 water-protein interactions to tailor the forcefield for αS. (**a**) Plot of LJ potentials with respect to $\lambda$. (**b**) The percentage bound values between two coarse-grained (CG) αS chains for different values of $\lambda$. The dashed black line represents the percentage bound values for two all-atom chains. (**c**) Error between $R_g$ calculated from CG and from all-atom simulations vs $\lambda$. The inset plot showcases the average values of $R_g$ obtained from CG along with their respective standard deviations. The dashed line represents the average value from all-atom simulations.

the CG model is not suited for simulations of disordered proteins and reported to underestimate the global dimensions of these systems in addition to overestimating protein-protein interactions. Previous attempts to simulate IDPs have modified the Martini force field by tuning the water-protein interactions, specifically, $\sigma$ and $\epsilon$ of Lennard-Jones interactions to render them suitable for modeling a specific IDP or all IDPs (**Benayad et al., 2021**; **Thomasen et al., 2022**; **Zerze, 2024**). Here, we follow a similar protocol, however instead of tuning only the $\epsilon$ part of the water-protein Lennard-Jones interactions, we refine both the $\sigma$ and $\epsilon$ parameters of the water-protein interactions (**Equation 9**).

$$V'(r) = 4\epsilon' \left[ (\frac{\sigma'}{r})^{12} - (\frac{\sigma'}{r})^6 \right]$$ (9)

where $\epsilon' = \lambda\epsilon$, $\sigma' = \lambda\sigma$, and $\lambda$ is the scaling parameter that needs to be optimized. Scaling $\sigma$ tunes the relative radius of the hydration spheres of each residue of a protein while a change in $\epsilon$ changes the strength of the water-residue interactions (**Figure 9a**). Increasing the $\epsilon$ value of water-protein interactions results in a higher energy demand for removing water molecules (dehydration) as a chain transitions from the dilute to the dense phase. Conversely, a higher $\sigma$ value implies that the hydration shell will be at a greater distance, facilitating dehydration if a chain moves into the dilute phase. Therefore, adjusting water-protein interactions based on the protein's single-chain behavior may not significantly influence the protein's phase behavior. Furthermore, fine-tuning both $\epsilon$ and $\sigma$ parameters only requires a minimal change in the overall protein-water interaction (1%). As a result, this adjustment minimally alters the force field parameters.

As we are interested in exploiting multi-chain simulations to study the LLPS of αS using Martini 3, we use the percentage of time two all-atom monomers remain bound to each other as the benchmark. To obtain an optimum scaling parameter for the water-protein interactions in Martini 3, specific to αS, we perform CG simulations with two αS chains with different values of $\lambda$. We start with two chains, without any secondary structure enforced upon them, randomly placed in a 15.7 nm box making sure that they are apart by at least 0.8 nm which we use as cutoff to classify the chains to be bound. If the minimum distance between any two residues belonging to the different chains are closer than 0.8 nm we consider them to be bound. Using the cutoff defined, we calculate the percentage bound between the two αS monomers for different values of $\lambda$ in the CG model. We also calculate the same from atomistic simulations reported in **Menon and Mondal, 2023**, as the reference. From **Figure 9b**, we can see that for multiple values of $\lambda$, we observe a close agreement in percentage bound values between CG and atomistic simulations.

We conducted additional single-chain CG simulations of αS, varying the parameter $\lambda$, while refraining from imposing any secondary structure constraints. Subsequently, we compared the mean $R_g$ values derived from these CG simulations with the 73 μs all-atom trajectory, which replaced the previously published 30 μs all-atom trajectory in **Robustelli et al., 2018**, and was provided by DE Shaw Research. **Figure 9c** illustrates that, for $\lambda = 1.01$, the average $R_g$ in the CG simulations closely

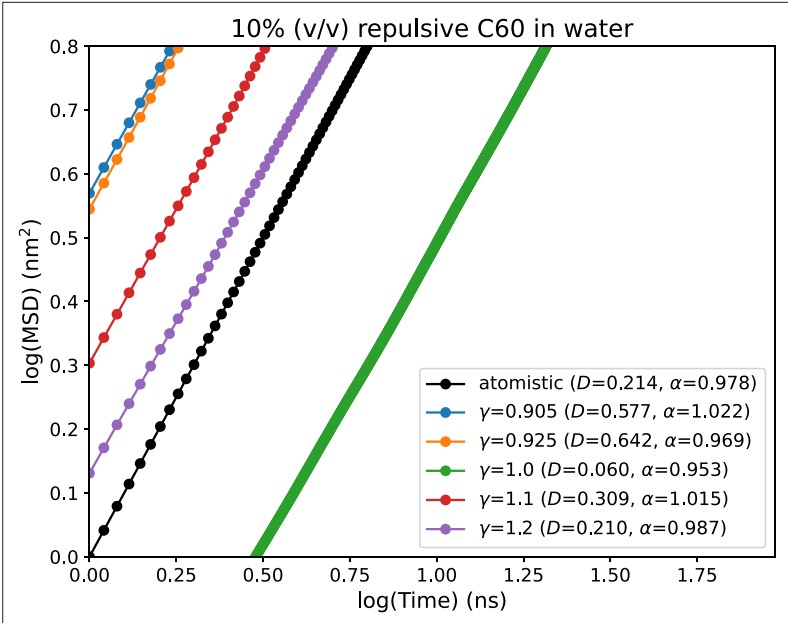

**Figure 10.** Mean squared displacements (MSD) vs time plots for different values of α. The black line represents the MSD obtained from atomistic simulations with purely repulsive fullerene-fullerene interactions.

matches the $R_g$ values obtained from the all-atom data. Consequently, we have chosen $\lambda = 1.01$ for the multi-chain simulations, as it minimizes errors for both single-chain $R_g$ and the observed percentage of time bound in the two-protein chain simulations.

## Porting fullerene-based crowder to Martini 3

In this study, we model the crowders as fullerenes that have purely repulsive interactions with each other. Their interactions are modeled as consisting of only the repulsive part of their Lennard-Jones interactions instead of the full potential (*Equation 10*).

$$V^{F-F}(r) = \frac{4\gamma\epsilon\sigma^{12}}{r} \tag{10}$$

where $V^{F-F}(r)$ is the interaction among different fullerenes and $\gamma = 1.0$ for the default parameters reported for Martini 2.

The parameters previously reported for fullerene is for Martini 2 CG force field. Therefore, we port the parameters first to Martini 3 by addition of new interactions in Martini 3 force field (CNP beads). We test the validity of the ported parameters of fullerene by calculating and comparing their mean squared displacements (MSD) with those obtained from atomistic simulations (see *Figure 10*). For this, we performed atomistic simulation of 10% (vol/vol) fullerene in water in a cubic box of ~5 nm as it is the concentration used with αS monomers as reported previously (*Menon and Mondal, 2023*). In a similar setup, we also run CG simulations of 10% (vol/vol) of fullerenes in water, where the volume of each fullerene-based crowder has been set to 0.55 nm³ (*Adams et al., 1994*). As shown in *Figure 10*, the default ported parameters of fullerene do not reproduce the MSD obtained in atomistic simulations. This indicates that the fullerene parameters need to be tuned to obtain a good agreement in this dynamical property (MSD). To achieve this, similar to the previous approach taken for modeling of αS in Martini 3, we tune the water-CNP interactions in Martini 3 (*Equation 10*). We iteratively vary $\gamma$ to match the MSD from CG simulations to the reference atomistic one. We observe that at $\gamma = 1.2$, we obtain the closest match between Martini 3 CG and atomistic simulations (*Figure 10*).

## Initial conformation generation for large-scale multi-chain simulations

A recent study used Markov state models to delineate the metastable states based on the extent of compaction ($R_g$) and identified three macrostates and their relative populations (*Menon and Mondal,*

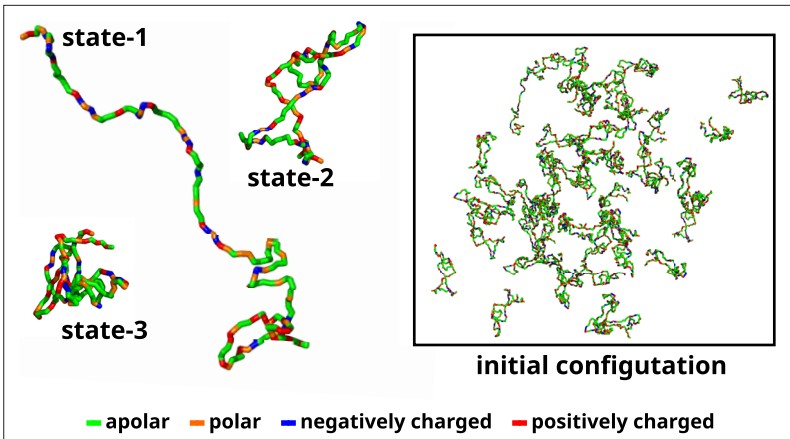

**Figure 11.** Initial configuration of αS for all multi-chain CG simulations. The left side of the figure shows the coarse-grained representation of three different conformation of αS. State-1 is the most extended conformation, followed by state-2 and finally state-3 which is the most compact conformation. The right side of the figure shows the mixture of all these conformations with a total of 50 chains in a cubic box. The residues have been color coded on the basis of their polarity/charge.

*2023*). Subsequent to the investigation, we utilize three representative conformations, each corresponding to one of the macrostates. We designate these macrostates as 1 (ms1), 2 (ms2), and 3 (ms3) (*Figure 11*). Therefore, in the multi-chain simulations, we maintain similar relative populations of these macrostates (Appendix 1—figure 6). The reported percentages of macrostates (labeled as ms1, ms2, and ms3) are 0.06%, 85.9%, and 14%, respectively. We added 50 αS monomers consisting of 1 chain of ms1, 45 chains of ms2, and 4 chains of ms3 in a cubic box with their respective secondary structures, determined via DSSP (*Kabsch and Sander, 1983*; *Joosten et al., 2011*; *Touw et al., 2015*), enforced using Martini 3. The size of the box of side $a$ is determined as per *Equation 11*.

$$a = \sqrt[3]{\frac{N}{N_A \times C}} \tag{11}$$

where $N$ is the number of monomers, $N_A$ is the Avogadro's number, and $C$ is the required concentration of αS.

Here, we simulate multiple concentrations of the protein, namely, 300, 400, 500, and 750 µM. 300 and 400 µM are below the critical concentrations required to undergo LLPS (*Ray et al., 2020*). We then solvate the system in CG water. We set up the 50-chain system to simulate three conditions: (i) in pure water, (ii) in 50 mM NaCl, and (iii) in presence of 10% (vol/vol) crowders. To study the effect of salt, we add the required number of $Na^+$ and $Cl^-$ ions to attain the desired concentration of 50 mM while also adding a few ions to render the system electrically neutral. In the system with crowders, we first add crowders after solvation by replacing a few solvent molecules with the required number of crowder molecules. We next resolvate the system along with the crowders. Finally we render the

**Table 1.** Details of the systems that were explored.

| Summary | Conc. of αS (µM) | Box size (nm) | # water | # crowders | # Na⁺ | # Cl⁻ |
|---|---|---|---|---|---|---|
| 300 µM αS in water | 300 | 65.66 | 2,304,122 | 0 | 450 | 0 |
| 400 µM αS in water | 400 | 59.69 | 1,729,213 | 0 | 450 | 0 |
| 500 µM αS in water | 500 | 55.52 | 1,389,721 | 0 | 450 | 0 |
| 750 µM αS in water | 750 | 48.42 | 920,023 | 0 | 450 | 0 |
| 750 µM αS + 10%(vol/vol) crowder | 750 | 48.42 | 843,011 | 20,128 | 450 | 0 |
| 750 µM αS + 50 mM NaCl | 750 | 48.42 | 913,357 | 0 | 3783 | 3333 |

For all simulations, a total of 50 monomeric protein chains have been used which comprise 1×ms1, 45×ms2, and 4×ms3.

**Table 2.** Runtimes of different simulations.

| System | No. of replicas | Runtime (s) |
| --- | --- | --- |
| 300 µM αS | 1 | 2.5 µs |
| 400 µM αS | 1 | 4.3 µs |
| 500 µM αS | 1 | 4.1 µs |
| 750 µM αS | 4 | 2.6, 3.1, 3.0, 3.5 µs |
| 750 µM αS + 10% (vol/vol) crowders | 4 | 2.8, 2.5, 2.6, 2.6 µs |
| 750 µM αS + 50 mM NaCl | 4 | 2.6, 2.4, 2.6, 2.3 µs |

system electro-neutral by addition of the required number of Na⁺ or Cl⁻ ions. The details of the simulation setup are provided in *Table 1*.

## Simulation setup

Upon successful generation of the initial conformation, we first perform an energy minimization using steepest gradient descent using an energy tolerance of 10 kJ/mol/nm. We next perform NVT simulations at 310.15 K using v-rescale thermostat for 5 ns using 0.01 ps as the time step. It is then followed by NPT simulation at 310.15 K and 1 bar using v-rescale thermostat and Berendsen barostat for 5 ns with a time step of 0.02 ps.

Next we perform CGMD simulations using velocity-verlet integrator with a time step of 0.02 ps using v-rescale thermostat at 310.15 K and Berendsen barostat at 1 bar. Both Lennard-Jones and electrostatic interactions are cut off at 1.1 nm. Coulombic interactions are calculated using reaction-field algorithm and relative dielectric constant of 15. We perform CGMD for at least 2.5 µs for the systems with 50 αS monomers. The details of the simulation runtimes have been provided in *Table 2*. We use the last 1 µs for further analyses.

## Ascertaining the attainment of steady state in simulation

In this study, we utilized the final 1 µs from each simulation for further analysis. To ascertain whether the simulations have achieved a steady state, we plotted the time profile of protein concentration in the dilute phase for all three cases.

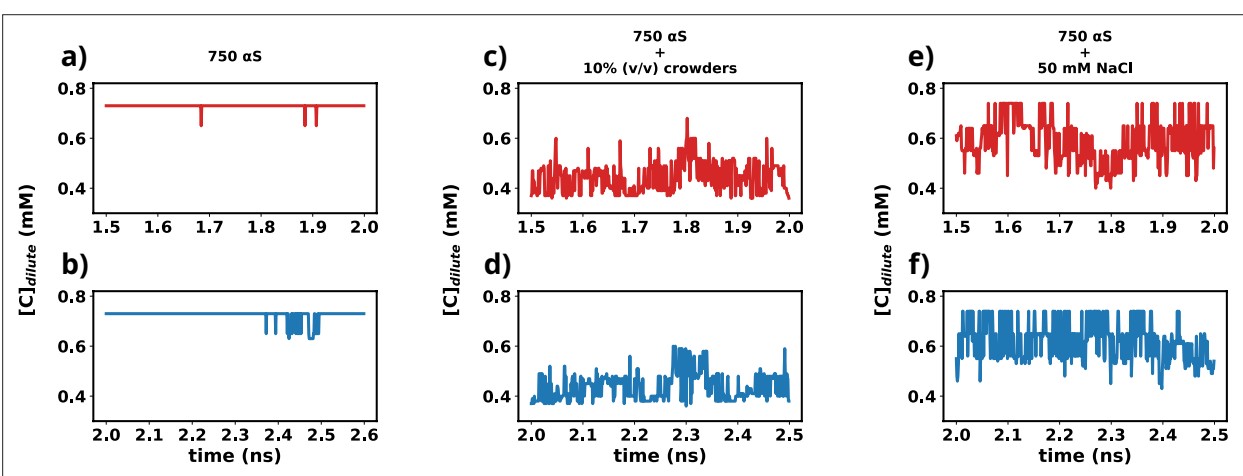

**Figure 12.** Time profiles of different metrics that showcase the attainment of steady state. (**a**) Concentration vs time profile of αS between 1.5 and 2.0 µs. (**b**) Concentration vs time profile of αS between 2.0 and 2.5 µs. (**c**) Concentration vs time profile of αS + 10% (vol/vol) crowders between 1.5 and 2.0 µs. (**d**) Concentration vs time profile of αS + 10% (vol/vol) crowders between 2.0 and 2.5 µs. (**e**) Concentration vs time profile of αS + 50 mM NaCl between 1.5 and 2.0 µs. (**f**) Concentration vs time profile of αS + 50 mM NaCl between 2.0 and 2.5 µs.

The online version of this article includes the following figure supplement(s) for figure 12:

**Figure supplement 1.** Assesment of equilibration of simulated trajectories.

Except for minor intermittent fluctuation involving only αS in neat water (**Figure 12a and b**), the remaining cases exhibit notably stable concentrations throughout various segments of the trajectory (**Figure 12c–f**). The relatively higher fluctuations observed in **Figure 12a and b** stem from the slow, spontaneous aggregation of αS alone, compounded by the inherently ambiguous nature of the dense phase. Consequently, the addition or removal of a few chains from the dense to the dilute phase results in significant fluctuations in protein concentration within the dilute phase. Conversely, in the other two scenarios (**Figure 12c–f**), aggregation is expedited by the presence of crowders/salt, leading to the formation of larger aggregates. Consequently, the addition or removal of one or two chains has negligible impact on concentration, thereby mitigating sudden large jumps. In summary, the conspicuous jumps depicted in **Figure 12a and b** arise from the gradual, fluctuating aggregation of pure αS and finite size effects. However, since these remain within the realm of fluctuations, we assert that the systems have indeed reached steady states. This assertion is bolstered by the subsequent figure, where the time profile of several pertinent system-wide macroscopic properties reveals no discernible change between 1.5 and 2.5 µs (**Figure 12—figure supplement 1**).

## Calculation of concentration of phases

We quantify the concentration of the protein in the solution (dilute phase) and in the aggregate (high-density phase) similar to the approach taken by **Nguyen et al., 2022**. We first calculate the volume of aggregate by **Equation 12** as follows:

$$V_{aggr} = \frac{4\pi}{3} \sqrt[3]{\lambda_1 \lambda_2 \lambda_3} \tag{12}$$

where $\lambda_1$, $\lambda_2$, and $\lambda_3$ are the eigenvalues from the gyration tensor of the aggregate. The concentration of the proteins in the aggregate is then calculated by **Equation 13**.

$$C_{aggr} = \frac{N}{N_A . V_{aggr}} \tag{13}$$

where $N$ is the number of chains in the aggregate, $N_A$ is the Avogadro's number, and $V_{aggr}$ is the volume of the aggregate obtained using **Equation 14**.

The concentration of the dilute phase is then calculated by **Equation 14**.

$$C_{dilute} = \frac{N_{dilute}}{N_A . V_{dilute}} \tag{14}$$

where $N_{dilute}$ is the number of chains present in trimers or lower aggregates, $N_A$ is the Avogadro's number, and $V_{dilute} = V_{system} - \sum_i V_{aggr}^i$. $\sum_i V_{aggr}^i$ is the total volume occupied by larger aggregates (6 or more).

## Calculation of small-worldness (*S*)

It is defined as per **Equation 15** (**Humphries and Gurney, 2008**).

$$S = \frac{\gamma_G}{\lambda_G} \tag{15}$$

where $\gamma_G = \frac{C_G^{\Delta}}{C_{rand}^{\Delta}}$ and $\lambda_G = \frac{L_G}{L_{rand}}$. $C_G$ is the mean clustering coefficient for a graph $G$ and $L_G$ is the mean shortest path length for $G$. $C_{rand}^{\Delta}$ and $L_{rand}$ are the mean clustering coefficient and mean shortest path length for an ensemble of Erdos-Renyi random network of the same size, respectively.

## Calculation of surface tension

We follow the procedure reported by **Benayad et al., 2021**, with only one minor difference. In **Benayad et al., 2021**, exact masses for the Martini beads were not taken into account, rather all beads were assumed to have the same mass. In this study we use the actual masses of the beads for all calculations.

As per **Benayad et al., 2021**, we first calculate the droplet shape using **Equations 16 and 17**.

$$C_{\alpha,\beta} = \frac{m_i(r_i^\alpha - r_{CMS}^\alpha)(r_i^\beta - r_{CMS}^\beta)}{\sum_i m_i} \tag{16}$$

where $C$ is the mass weighted covariance matrix, $\alpha$ and $\beta$ are directions $x, y$ or $z$; $i$ is the index for atoms/beads of protein monomers within a droplet, $r_{CMS}^{\alpha/\beta}$ is the center of mass of the droplet in $x, y$, or $z$ direction; $m_i$ is the mass of the atom/bead.

The eigenvalues $\lambda_1, d_2$, and $\lambda_3$ of $C$ are given by: $\lambda_1 = vh^2$, $\lambda_2 = vb^2$, $\lambda_3 = vc^2$. Since $R^3 = abc$, where $R$ is the average droplet radius, we obtain *Equation 17*:

$$a = \frac{R\lambda_1^{1/3}}{(\lambda_2\lambda_3)^{1/6}}$$

$$b = \frac{R\lambda_2^{1/3}}{(\lambda_1\lambda_9)^{1/6}}$$

$$c = \frac{R\lambda_3^{1/3}}{(\lambda_1\lambda_2)^{1/6}} \tag{17}$$

We next define $\delta a = a - R$, $\delta b = b - R$, and $\delta c = c - R$. Using these, we obtain *Equation 18*:

$$\left\langle (\delta a \pm \delta b)^2 \right\rangle = \frac{1}{3} \sum_{i=1}^{2} \sum_{j=i+1}^{3} \left\langle (\delta a_i \pm \delta a_j)^2 \right\rangle \tag{18}$$

Therefore, the surface tension ($\gamma$) is then estimated using $\gamma \approx \gamma_{20} \approx \gamma_{22}$, where

$$\gamma_{20} = \frac{5k_BT}{16\pi\langle(\delta a - \delta b)^2\rangle}$$

$$\gamma_{20} = \frac{5k_BT}{16\pi\langle(\delta a - \delta b)^2\rangle} \tag{19}$$

## List of software

We have used only open-source software for this study. All simulations have been performed using GROMACS-2021 (*Van Der Spoel et al., 2005*; *Abraham et al., 2015*). Snapshots were generated using PyMOL 2.5.4 (*Schrödinger, 2015a*; *Schrödinger, 2015b*; *Schrödinger, 2015c*). Analysis were performed using Python (*Van Rossum and Drake, 2009*) and MDAnalysis (*Michaud-Agrawal et al., 2011*; *Gowers et al., 2016*). Figures were prepared using Matplotlib (*Hunter, 2007*), Jupyter (*Kluyver et al., 2016*), and (*Inkscape, 2024*).

## Acknowledgements

We acknowledge support of the Department of Atomic Energy, Government of India, under Project Identification No. RTI 4007. We also acknowledge Core Research grants provided by the Department of Science and Technology (DST) of India (CRG/2023/001426).

## Additional information

### Funding

| Funder | Grant reference number | Author |
|---|---|---|
| Department of Atomic Energy, Government of India | RTI 4007 | Abdul Wasim Sneha Menon Jagannath Mondal |
| Department of Science and Technology, Ministry of Science and Technology, India | CRG/2023/001426 | Jagannath Mondal |

| Funder | Grant reference number | Author |
|--------|------------------------|--------|

The funders had no role in study design, data collection and interpretation, or the decision to submit the work for publication.

## Author contributions

Abdul Wasim, Conceptualization, Data curation, Formal analysis, Validation, Investigation, Methodology, Writing - original draft, Writing - review and editing; Sneha Menon, Formal analysis, Writing - original draft; Jagannath Mondal, Conceptualization, Supervision, Funding acquisition, Validation, Writing - original draft, Project administration, Writing - review and editing

## Author ORCIDs

Jagannath Mondal (iD) https://orcid.org/0000-0003-1090-5199

Reviewer #1 (Public Review): https://doi.org/10.7554/eLife.95180.3.sa1
Reviewer #2 (Public Review): https://doi.org/10.7554/eLife.95180.3.sa2
Author response https://doi.org/10.7554/eLife.95180.3.sa3

---

## Additional files

### Supplementary files

• MDAR checklist

### Data availability

All data are present within the manuscript. The sections of the trajectories used for analysis (1.5 - 2.0 μs) have been uploaded to zenodo (https://doi.org/10.5281/zenodo.10926367).Other relevant files related to the simulations have also been uploaded to the same repository.

The following dataset was generated:

| Author(s) | Year | Dataset title | Dataset URL | Database and Identifier |
|-----------|------|---------------|-------------|-------------------------|
| Wasim A, Sneha M, Jagannath M | 2024 | Modulation of α-Synuclein Aggregation Amid Diverse Environmental Perturbation | https://doi.org/10.5281/zenodo.10926367 | Zenodo, 10.5281/zenodo.10926367 |

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
