## [Editor Report · eLife assessment]

This study provides **important** biophysical insights into the molecular mechanism underlying the association of alpha-synuclein chains, which is essential for understanding the pathogenesis of Parkinson's disease. The data analysis is **solid**, and the methodology can help investigate other molecular processes involving intrinsically disordered proteins.

---

## [Referee Report · Reviewer #1 (Public Review)]

Summary:

In this paper, the authors performed molecular dynamics (MD) simulations to investigate the molecular basis of association of alpha-synuclein chains under molecular crowding and salt conditions. Aggregation of alpha-synuclein is linked to the pathogenesis of Parkinson's disease, and the liquid-liquid phase separation (LLPS) is considered to play an important role in the nucleation step of the alpha-synuclein aggregation. This paper re-tuned the Martini3 coarse-grained force field parameters, which allows long-timescale MD simulations of intrinsically disordered proteins with explicit solvent under diverse environmental perturbation. Their MD simulations showed that alpha-synuclein does not have a high LLPS-forming propensity, but the molecular crowding and salt addition tend to enhance the tendency of droplet formation and therefore modulate the alpha-synuclein aggregation. The MD simulation results also revealed important intra and inter-molecule conformational features of the alpha-synuclein chains in the formed droplets and the key interactions responsible for the stability of the droplets. These MD simulation data add biophysical insights into the molecular mechanism underlying the association of alpha-synuclein chains, which may be useful for understanding the pathogenesis of Parkinson's disease.

Strengths:

(1) The re-parameterized Martini 3 coarse-grained force field enables the large-scale MD simulations of the intrinsically disordered proteins with explicit solvent, which will be useful for a more realistic description of the molecular basis of LLPS.

(2) This paper showed that the molecular crowding and salt contribute to the modulation of the LLPS through different means. The molecular crowding minimally affects surface tension, but adding salt increases surface tension. It is also interesting to show that the aggregation pathway involves the disruption of the intra-chain interactions arising from C-terminal regions, which potentially facilitates the formation of inter-chain interactions.

Weaknesses:

(1) Although the authors emphasized the advantage of the Martini3 force field for its explicit description of solvent, this paper did not analyze the water behavior contained in the simulation trajectories and discuss the water's role in the aggregation and LLPS.

(2) This paper discussed the effects of crowders and salt on the surface tension of the droplets. The calculation of the surface tension relies on the droplet shape. However, for the formed clusters in the MD simulations, the typical size is <10, which may be too small to rigorously define the droplet shape. As shown in previous work cited by this paper [Benayad et al., J. Chem. Theory Comput. 2021, 17, 525−537], the calculated surface tension becomes stable when the chain number is larger than 100.

(3) Both the sizes and volume fractions of the crowders can affect the protein association. It will be interesting to perform MD simulations by adding crowders with various sizes and volume fractions. In addition, in this work the crowders were modelled by fullerenes, which contribute to protein aggregation mainly by entropic means as discussed in the manuscript. It is not very clear how the crowder effect is sensitive to the chemical nature of the crowders (e.g., inert crowers with excluded volume effect or crowders with non-specific attractive interactions with proteins, etc).

---

## [Referee Report · Reviewer #2 (Public Review)]

In the manuscript "Modulation of α-Synuclein Aggregation Amid Diverse Environmental Perturbation", Wasim et al describe coarse-grained molecular dynamics (cgMD) simulations of α-Synuclein (aSyn) at several concentrations and in the presence of molecular crowding agents or high salt. They begin by bench-marking their cgMD against all-atom simulations by Shaw. They then carry 2.4-4.3 µs cgMD simulations under the above-noted conditions and analyze the data in terms of protein structure, interaction network analysis, and extrapolated fluid mechanics properties. This is an interesting study because a molecular scale understanding of protein droplets is currently lacking.

---

## [Author Response]

The following is the authors’ response to the original reviews.

**Reviewer #1**
Summary:In this paper, the authors performed molecular dynamics (MD) simulations to investigate the molecular basis of the association of alpha-synuclein chains under molecular crowding and salt conditions. Aggregation of alpha-synuclein is linked to the pathogenesis of Parkinson's disease, and the liquid-liquid phase separation (LLPS) is considered to play an important role in the nucleation step of the alpha-synuclein aggregation. This paper re-tuned the Martini3 coarse-grained force field parameters, which allows long-timescale MD simulations of intrinsically disordered proteins with explicit solvent under diverse environmental perturbation. Their MD simulations showed that alpha-synuclein does not have a high LLPS-forming propensity, but the molecular crowding and salt addition tend to enhance the tendency of droplet formation and therefore modulate the alpha-synuclein aggregation. The MD simulation results also revealed important intra- and inter-molecule conformational features of the alpha-synuclein chains in the formed droplets and the key interactions responsible for the stability of the droplets. These MD simulation data add biophysical insights into the molecular mechanism underlying the association of alpha-synuclein chains, which is important for understanding the pathogenesis of Parkinson's disease.Strengths:(1) The re-parameterized Martini 3 coarse-grained force field enables the large-scale MD simulations of the intrinsically disordered proteins with explicit solvent, which will be useful for a more realistic description of the molecular basis of LLPS.(2) This paper showed that molecular crowding and salt contribute to the modulation of the LLPS through different means. The molecular crowding minimally affects surface tension, but adding salt increases surface tension. It is also interesting to show that the aggregation pathway involves the disruption of the intra-chain interactions arising from C-terminal regions, which potentially facilitates the formation of inter-chain interactions.

We thank the reviewer for pointing out the strengths of our study.

Weaknesses:(1) Although the authors emphasized the advantage of the Martini3 force field for its explicit description of solvent, the whole paper did not discuss the water's role in the aggregation and LLPS.

We thank the reviewer for pointing this out. We agree that we have not explored or discussed the role of water in aS aggregation or LLPS. We would like to convey that we would like to explore that in detail in a separate study altogether. However we have updated the “Discussion” section with the following lines to convey to the readers the importance water plays in aggregation and LLPS of aS.

Page 24: “The significance of the solvent in alpha-synuclein (αS) aggregation remains underexplored. Recent studies [26, 55] underscore the pivotal role of water as a solvent in LLPS. It suggests that comprehending the solvent’s role, particularly water, is essential for attaining a deeper grasp of the thermodynamic and physical aspects of αS LLPS and aggregation. By delving into the solvent’s contribution, researchers can uncover additional factors influencing αS aggregation. Such insights hold the potential to advance our comprehension of protein aggregation phenomena, crucial for devising strategies to address diseases linked to protein misfolding and aggregation, notably Parkinson’s disease. Future investigations focusing on elucidating the interplay between αS, solvent (especially water), and other environmental elements could yield valuable insights into the mechanisms underlying LLPS and aggregation. Ultimately, this could aid in the development of therapeutic interventions or preventive measures for Parkinson’s and related diseases.”

(2) This paper discussed the effects of crowders and salt on the surface tension of the droplets.The calculation of the surface tension relies on the droplet shape. However, for the formed clusters in the MD simulations, the typical size is <10, which may be too small to rigorously define the droplet shape. As shown in previous work cited by this paper [Benayad et al., J. Chem. Theory Comput. 2021, 17, 525−537], the calculated surface tension becomes stable when the chain number is larger than 100.

We appreciate the insightful feedback from the reviewer. However, we would like to emphasize that the αS droplets exhibit a highly liquid-like behavior, characterized by frequent exchanges of chains between the dense and dilute phases, alongside a slow aggregation process. In the study by Benayad et al. (2020, JCTC) [ref. 30], FUS-LCD was the protein of choice at concentrations in the (mM) range. FUS-LCD is known to undergo very rapid LLPS at concentrations lower than 100 (μM) where for αS the critical concentration for LLPS is 500 (μM) and undergoes slower aggregation than FUS. Moreover, the diffusion constant of αS inside newly formed droplets (no liquid to solid phase transition has occurred) has been estimated to be 0.23-0.58 μm2/s (Ray et al, 2020, Nat. Comm.). The value of diffusion constant for FUS-LCD inside LLPS droplets has been estimated to be 0.17 μm2/s (Murthy et al. 2023, Nat. Struct. and Mol. Biol.). These prove that αS forms droplets that are less viscous than that formed by FUS-LCD. This dynamic nature impedes the formation of large droplets in the simulations, making it challenging to rigorously calculate surface tension from interfacial width, which, in turn, necessitates the computation of *g(r)* between water and the droplet.

Furthermore, it's essential to note that our primary aim in calculating surface tension was not to determine its absolute value. Rather, we aimed to compare surface tensions obtained for the three distinct environments explored in this study. Hence, our primary objective is to compare the distributions of surface tensions rather than focusing solely on the mean values obtained. The distributions shown in Figure 4a clearly show a trend which we have stated in the article.

(3) In this work, the Martini 3 force field was modified by rescaling the LJ parameters \epsilon and \sigma with a common factor \lambda. It has not been very clearly described in the manuscript why these two different parameters can be rescaled by a common factor and why it is necessary to separately tune these two parameters, instead of just tuning the coefficient \epsilon as did in a previous work [Larsen et al., PLoS Comput Biol 16: e1007870].

We thank the reviewer for the comment. We think that the distance of the first hydration layer also should have an impact on aggregation/LLPS. Here we are scaling both the epsilon and sigma. A higher epsilon of water-protein interactions mean higher the energy required for removal of water molecules (dehydration) when a chain goes from the dilute to the dense phase. A higher sigma on the other hand means that the hydration shell will also be at a larger distance making dehydration easier. Moreover, tuning both (either by same or different parameter) required a change of the overall protein-water interaction by only 1%, thereby requiring only considerably minimal change in forcefield parameters (compared to the case where only epsilon is being tuned which required 6-10% change in epsilon from its original values.) . Thus we think one of the ways of tuning water-protein interactions which requires minimal retuning of Martini 3 is by optimizing both epsilon and sigma. However whether a single scaling parameter is good enough requires further exploration and is outside the scope of the current study. More importantly it would introduce another free parameter into the system and the lesser the number of free parameters, the better. For this study, a single parameter sufficed as depicted in Figure 9. To inform the readers of why we chose to scale both sigma and epsilon, we have added the following in the main text:

Page 25-26: “Increasing the ϵ value of water-protein interactions results in a higher energy demand for removing water molecules (dehydration) as a chain transitions from the dilute to the dense phase. Conversely, a higher σ value implies that the hydration shell will be at a greater distance, facilitating dehydration if a chain moves into the dilute phase. Therefore, adjusting water-protein interactions based on the protein’s single-chain behavior may not significantly influence the protein’s phase behavior. Furthermore, fine-tuning both ϵ and σ parameters only requires a minimal change in the overall protein-water interaction (1%). As a result, this adjustment minimally alters the force field parameters.”

(4) Both the sizes and volume fractions of the crowders can affect the protein association. It will be interesting to perform MD simulations by adding crowders with various sizes and volume fractions. In addition, in this work, the crowders were modelled by fullerenes, which contribute to protein aggregation mainly by entropic means as discussed in the manuscript. It is not very clear how the crowder effect is sensitive to the chemical nature of the crowders (e.g., inert crowders with excluded volume effect or crowders with non-specific attractive interactions with proteins, etc) and therefore the force field parameters.

We thank the reviewer for a potential future direction. In this investigation our main focus was to simulate the inertness features of crowders only, to ensure that only entropic effect of the crowders are explored. Although this study focuses on the factors that enable aS to form an aggregates/LLPS under different environmental conditions, it would be interesting to explore in a systematic way the mechanism of action of crowders of varying shapes, sizes and interactions. Therefore we added the following lines in the “Discussion” section to let the readers know that this is also a future prospect of investigation.

Page 22: “Under physiological conditions, crowding effects emerge prominently. While crowders are commonly perceived to be inert, as has been considered in this investigation, the morphology, dimensions, and chemical interactions of crowding agents with αS in both dilute and dense phases may potentially exert considerable influence on its LLPS. Hence, a comprehensive understanding through systematic exploration is another avenue that warrants extensive investigation.”

**Reviewer #1 (Recommendations For The Authors):**
(1) Figure S1. The title of the figure and the description in the figure caption are inconsistent?

We thank the reviewer for the comment and we have updated the article with the correct caption.

(2) Page 14, line 3, the authors may want to provide more descriptions of the "ms1", "ms2", and "ms3" for better understanding.

We are grateful to the reviewer for pointing this out. We have added a line describing in brief what “ms1”, “ms2” and “ms3” represent. It reads “Subsequent to the investigation, we utilize three representative conformations, each corresponding to one of the macrostates. We designate these macrostates as 1 (ms1), 2 (ms2), and 3 (ms3) (Figure S7)” (Page 28)

(3) Page 20, the authors may want to briefly explain how the normalized Shannon entropy was calculated.

We thank the reviewer for pointing this out. This is plain Shannon Entropy and the word “normalized” should not have been there. To avoid confusion we have provided the equation we have used to calculate the Shannon entropy (Eq 8) (Page 21).

**Reviewer #2 (Public Review):**
In the manuscript "Modulation of α-Synuclein Aggregation Amid Diverse Environmental Perturbation", Wasim et al describe coarse-grained molecular dynamics (cgMD) simulations of α-Synuclein (αS) at several concentrations and in the presence of molecular crowding agents or high salt. They begin by bench-marking their cgMD against all-atom simulations by Shaw. They then carry 2.4-4.3 µs cgMD simulations under the above-noted conditions and analyze the data in terms of protein structure, interaction network analysis, and extrapolated fluid mechanics properties. This is an interesting study because a molecular scale understanding of protein droplets is currently lacking, but I have a number of concerns about how it is currently executed and presented.

We thank the reviewer for finding our study interesting.

(1) It is not clear whether the simulations have reached a steady state. If they have not, it invalidates many of their analysis methods and conclusions.

We have used the last 1 μs (1.5-2.5 1 μs) from each simulation for further analysis in this study. To understand whether the simulations have reached steady state or not, we plot the time profile of the concentration of the protein in the dilute phase for all three cases.

Except for the scenario of only αS (Figures a and b), the rest show very steady concentrations across various sections of the trajectory (Figures c-f). The larger sudden fluctuations observed inFigures a and b are due to the fact that only αS undergo very slow spontaneous aggregation and owing to the fact that the dense phase itself is very fluxional, addition/removal of a few chains to/from the dense to dilute phase register themselves as large fluctuations in the protein concentration in the dilute phase. For the other two scenarios (Figures c-f) aggregation has been accelerated due to the presence of crowders/salt. This causes larger aggregates to be formed. Therefore addition/removal of one or two chains does not significantly affect the concentration and we do not see such sudden large jumps. In summary, the large jumps seen in Figures a and b are due to slow, fluxional aggregation of pure αS and finite size effects. However as these still are only fluctuations, we posit that the systems have reached steady states. This claim is further supported by the following figure where the time profile of a few useful system wide macroscopic properties show no change between 1.5-2.5 µs.

We also have added a brief discussion in the Methods section (Page 29-30) with these figures in the Supplementary Information.

**Author response image 2. sa3fig2:** 

“In this study, we utilized the final 1 µs from each simulation for further analysis. To ascertain whether the simulations have achieved a steady state, we plotted the time profile of protein concentration in the dilute phase for all three cases. Except for minor intermittent fluctuation involving only αS in neat water (Figures S8a and S8b), the remaining cases exhibit notably stable concentrations throughout various segments of the trajectory (Figures S8 c-f). The relatively higher fluctuations observed in Figures S8a and b stem from the slow, spontaneous aggregation of αS alone, compounded by the inherently ambiguous nature of the dense phase.

Consequently, the addition or removal of a few chains from the dense to the dilute phase results in significant fluctuations in protein concentration within the dilute phase. Conversely, in the other two scenarios (Figures S8c-f), aggregation is expedited by the presence of crowders/salt, leading to the formation of larger aggregates. Consequently, the addition or removal of one or two chains has negligible impact on concentration, thereby mitigating sudden large jumps. In summary, the conspicuous jumps depicted in Figures S8a and b arise from the gradual, fluctuating aggregation of pure αS and finite size effects. However, since these remain within the realm of fluctuations, we assert that the systems have indeed reached steady states. This assertion is bolstered by the subsequent figure, where the time profile of several pertinent system-wide macroscopic properties reveals no discernible change between 1.5-2.5 µs (Figures S9).”

(2) The benchmarking used to validate their cgMD methods is very minimal and fails to utilize a large amount of available all-atom simulation and experimental data.

We disagree with the reviewer on this point. We have cited multiple previous studies [26, 27] that have chosen Rg as a metric of choice for benchmarking coarse-grained model and have used a reference (experimental or otherwise) to tune Martini force fields. Majority of the notable literature where Rg was used as a benchmark during generation of new coarse-grained force fields are works by Dignon et al. (PLoS Comp. Biol.) [ref. 25], Regy et al (Protein Science. 2021) [ref. 26], Joseph et al.(Nature Computational Science. 2021) [ref. 27] and Tesei et al (Open Research Europe, 2022) [ref. 28]. From a polymer physics perspective, tuning water-protein interactions is simply changing the solvent characteristics for the biopolymer and Rg has been generally considered a suitable metric in the case of coarse-grained model. Moreover we try to match the distribution of the Rg rather than only the mean value. This suggests that at a single molecule level, the cgMD simulations at the optimum water of water-protein interactions would allow the protein to sample the conformations present in the reference ensemble. We use the extensively sampled 70 μs all-atom data from DE Shaw Research to obtain the reference Rg distribution. Also we perform a cross validation by comparing the fraction of bound states in all-atom and cgMD dimer simulations which also seem to corroborate well with each other at optimum water-protein interactions. To let the readers understand the rationale behind choosing Rg we have added a section in the Methods section (Page 25) that explains why Rg is plausibly a good metric for tuning water-protein interactions in Martini 3, at least when dealing with IDPs.

Our optimized model is further supported by the FRET experiments by Ray et al. [6]. They found that interchain NAC-NAC interactions drive LLPS. Residue level contact maps obtained from our simulations also show decreased intrachain NAC-NAC interactions with an increased interchain NAC-NAC interactions inside the droplet. This corroborates well with the experimental observations and furthermore validates the metrics we have used for optimization of the water-protein interactions. However the comparison with the FRET data by Ray et al. was not present earlier and we have added the following lines in the updated draft.

Page17: “Thus we observed that increased inter-chain NAC-NAC regions facilitate the formation of αS droplets which also have previously been seen from FRET experiments on αS LLPS

droplets[6].”

(3) They also miss opportunities to compare their simulations to experimental data on aSyn protein droplets.

We thank the reviewer for pointing this out. We have tried to compare the results from our simulations to existing experimental FRET data on αS. Please see the previous response where we have described our comparison with FRET observations.

(4) Aspects such as network analysis are not contextualized by comparison to other protein condensed phases.

For a proper comparison between other protein condensed phases, we would require the position phase space of such condensates which is not readily available. Therefore we tried to explain it in a simpler manner to paint a picture of how αS forms an interconnecting network inside the droplet phase.

(5) Data are not made available, which is an emerging standard in the field.

We thank the reviewer for mentioning this. We have provided the trajectories between 1.5-2.5 μs, which we used for the analysis presented in the article, via a zenodo repository along with other relevant files related to the simulations (https://zenodo.org/records/10926368).

Firstly, it is not clear that these systems are equilibrated or at a steady state (since protein droplets are not really equilibrium systems). The authors do not present any data showing time courses that indicate the system to be reaching a steady state. This is problematic for several of their data analysis procedures, but particularly in determining free energy of transfer between the condensed and dilute phases based on partitioning.

We have addressed this concern as stated previously in the response. We have updated the article accordingly.

Secondly, the benchmarking that they perform against the 73 µs all-atom simulation of aSyn monomer by Shaw and coworkers provides only very crude validation of their cgMD models based on reproducing Rg for the monomer. The authors should make more extensive comparisons to the specific conformations observed in the DE Shaw work. Shaw makes the entire trajectory publicly available. There are also a wealth of experimental data that could be used for validation with more molecular detail. See for example, NMR and FRET data used to benchmark Monte Carlo simulations of aSyn monomer (as well as extensive comparisons to the Shaw MD trajectory) in Ferrie at al: A Unified De Novo Approach for Predicting the Structures of Ordered and Disordered Proteins, J. Phys. Chem. B 124 5538-5548 (2020)DOI:10.1021/acs.jpcb.0c02924I note that NMR measurements of aSyn in liquid droplets are available from Vendruscolo: Observation of an α-synuclein liquid droplet state and its maturation into Lewy body-like assemblies, Journal of Molecular Cell Biology, Volume 13, Issue 4, April 2021, Pages 282-294, https://doi.org/10.1093/jmcb/mjaa075.In addition, there are FRET studies by Maji: Spectrally Resolved FRET Microscopy of α-Synuclein Phase-Separated Liquid Droplets, Methods Mol Biol 2023:2551:425-447. doi: 10.1007/978-1-0716-2597-2_27.So the authors are missing opportunities to better validate the simulations and place their structural understanding in greater context. This is just based on my own quick search, so I am sure that additional and possibly better experimental comparisons can be found.

We have performed a comparison with existing FRET measurements by Ray et al. (2020) as discussed in a previous response and also updated the same in the article. The doi (10.1007/978-1-0716-2597-2_27) provided by the reviewer is however for a book on Methods to characterize protein aggregates and does not contain any information regarding the observations from FRET experiments. The other doi (https://doi.org/10.1093/jmcb/mjaa075) for the article from Vendrusculo group does not contain information directly relevant to this study. Moreover NMR measurements cannot be predicted from cgMD since full atomic resolution is lost upon coarse-graining of the protein . A past literature survey by the authors found very little scientific literature on molecular level characterization of αS LLPS droplets.

Thirdly, the small word network analysis is interesting, but hard to contextualize. For instance, the 8 Å cutoff used seems arbitrary. How does changing the cutoff affect the value of S determined? Also, how does the value of S compare to other condensed phases like crystal packing or amyloid forms of aSyn?

The 8 Å cutoff is actually arbitrary since a distance based clustering always requires a cutoff which is empirically decided. However 8 Å is quite large compared to other cutoffs used for distance based clustering. For example in ref 26, 5 Å was used as a cutoff for calculation of protein clusters. Larger cutoffs will lead to sparser network structures. However we used the same cutoff for all distance based clustering which makes the networks obtained comparable. We wanted to perform a comparison among the networks formed by αS under different environmental conditions.

Fourthly, I see no statement on data availability. The emerging standard in the computational field is to make all data publicly available through Github or some similar mechanism.

We thank the reviewer for pointing this out and we have provided the raw data between 1.5-2.5 μs for each scenario along with other relevant files via a zenodo repository (https://zenodo.org/records/10926368).

Finally, on page 16, they discuss the interactions of aSyn(95-110), but the sequence that they give is too long (seeming to contain repeated characters, but also not accurate). aSyn(95-110) = VKKDQLGKNEEGAPQE. Presumably this is just a typo, but potentially raises concerns about the simulations (since without available data, one cannot check that the sequence is accurate) and data analysis elsewhere.

This indeed is a typographical error. We have updated the article with the correct sequence. The validity of the simulations can be verified from the data we have shared via the zenodo repository (https://zenodo.org/records/10926368).